# ExeDec: Execution Decomposition for Compositional Generalization in Neural Program Synthesis

**Kensen Shi**
Google DeepMind
kshi@google.com

**Joey Hong** [*]
UC Berkeley
joey_hong@berkeley.edu

**Yinlin Deng** [*]
University of Illinois
Urbana-Champaign
yinlind2@illinois.edu

**Pengcheng Yin**
Google DeepMind
pcyin@google.com

**Manzil Zaheer**
Google DeepMind
manzilzaheer@google.com

**Charles Sutton**
Google DeepMind
charlessutton@google.com

## Abstract

When writing programs, people have the ability to tackle a new complex task by decomposing it into smaller and more familiar subtasks. While it is difficult to measure whether neural program synthesis methods have similar capabilities, we can measure whether they compositionally generalize, that is, whether a model that has been trained on the simpler subtasks is subsequently able to solve more complex tasks. In this paper, we characterize several different forms of compositional generalization that are desirable in program synthesis, forming a meta-benchmark which we use to create generalization tasks for two popular datasets, RobustFill and DeepCoder. We then propose ExeDec, a novel decomposition-based synthesis strategy that predicts execution subgoals to solve problems step-by-step informed by program execution at each step. When used with Transformer models trained from scratch, ExeDec has better synthesis performance and greatly improved compositional generalization ability compared to baselines. Finally, we use our benchmarks to demonstrate that LLMs struggle to compositionally generalize when asked to do programming-by-example in a few-shot setting, but an ExeDec-style prompting approach can improve the generalization ability and overall performance.

## 1 Introduction

*Program synthesis* aims to assist programmers by automatically producing code according to a user's specification of what the code should do (Gulwani et al., 2017). Program synthesis systems, such as programming by example (PBE) systems, have been effective for tasks such as string manipulation (Gulwani, 2011; Devlin et al., 2017; Shi et al., 2022b), writing short Java functions (Shi et al., 2019), and tensor manipulation (Shi et al., 2022a). Neural program synthesizers, especially those based on large language models (Chen et al., 2021a; Austin et al., 2021; Li et al., 2022), have been particularly successful at generating code functions and blocks across a variety of general-purpose programming languages.

An essential capability of human programmers is their ability to generalize by recombining parts of prior knowledge to solve new tasks. For example, a capable programmer can quickly adapt to new concepts and APIs, and compose different code idioms in unseen ways to solve novel problems. These skills are instances of *compositional generalization*, which is the ability to generalize to test examples consisting of different compositions of components individually seen during training (Keysers et al., 2020). While compositionality has been studied in natural language processing (Chomsky, 1957; Lake & Baroni, 2018; Gu et al., 2021), it has not been studied deeply in the context of programming by example. This problem is potentially fruitful not only because it might help to build more robust program synthesizers, but also as an example of how more general problem-solving is compositional.

---

[*]These authors contributed during internships at Google DeepMind.

To build neural synthesis systems that are better at compositional generalization, we propose designing systems that learn to *decompose* a complex task into a list of simpler subtasks. Each subtask is defined by a goal, so the process of decomposing a task is essentially planning. Indeed, decomposition is a skill so fundamental to software engineering that the first programming course at Stanford University introduces decomposition within the first week (Parlante, 2022). This can enable compositional generalization because subtasks seen during training can be combined in different ways at test time.

Based on this intuition, we propose ExeDec, a novel search method for neural program synthesis that performs decomposition within the *execution space*. A PBE task defines a program by pairs of program inputs with their desired outputs. Thus, it is natural to describe a subgoal by the desired *intermediate state*, i.e., values of local variables, for the next subtask. To describe the intuition in another way, we imagine that a human programmer does not decide on what code to write one token at a time, but rather thinks about what the result of the next code block should be, and then writes code to accomplish that. Specifically, ExeDec uses two neural models, a *subgoal model* that predicts the desired program state for the next part of the program, and a *synthesizer model* that attempts to generate a program that reaches that subgoal from the prior state. We interleave neural prediction with program execution within a beam search that enables exploring different predicted decompositions.

To evaluate this approach, we introduce a new meta-benchmark for measuring the compositional generalization abilities of program synthesizers. Given a standard program synthesis benchmark containing a domain-specific language and a distribution over target programs, our meta-benchmark describes train-test splits for 5 different types of compositional generalization, such as length generalization or composing API functions in different combinations in the training and test sets. While ExeDec has slightly better performance than a Transformer baseline in the i.i.d. setting, ExeDec also achieves a $2\times$ to $4\times$ accuracy increase in the compositional generalization setting. Additionally, ExeDec improves upon an ablation that does not explicitly propose subgoals, showing the importance of reasoning about execution subgoals instead of directly predicting code.

Interestingly, a similar approach can be applied to explore compositional generalization in large language models (LLMs). We explore whether the LLM can solve PBE tasks that compositionally generalize beyond those in a few-shot prompt. We similarly find that the LLM performs significantly worse when compositional generalization is required, and that an adaptation of ExeDec to the few-shot prompting setup increases the LLM's performance overall, including in compositional generalization. Even so, compositional generalization during program generation in LLMs remains a challenge.

## 2 COMPOSITIONAL GENERALIZATION IN PROGRAMMING

The goal in program synthesis is to find a program in a given language that is consistent with a specification. Formally, we are given a domain specific language (DSL) which defines a set $\mathcal{P}$ of programs. Elements in the DSL include functions (which we call *operations*), identifiers, constants, and so on. In programming by example (PBE), the desired program is specified by a set of input/output (I/O) examples denoted $X = \{(I_1, O_1), \ldots (I_n, O_n)\}$. Then, solving specification $X$ means finding a program $P \in \mathcal{P}$ that correctly solves all of the examples: $P(I_i) = O_i, \ \forall i$. A robust program synthesizer should generalize to programs not in the training set. Regardless of the programming language or DSL, programs are nearly always built from smaller parts, which we call *subprograms*, such as lines and blocks of code, functions, and so on. For compositional generalization, we are interested in whether the synthesizer can combine subprograms in new ways from the training set.

We design our benchmark around five compositional generalization tasks applicable to program synthesis (Figure 1). These tasks measure whether synthesizers can generalize to longer programs or to programs that use *concepts*, such as API methods, in different compositional ways. These concepts partition the DSL operations into groups.[1] In this section, we describe the generalization tasks abstractly, forming a *meta-benchmark* that can be applied in future work to construct new compositional generalization benchmarks using existing datasets or DSLs. Then, in Section 3, we concretize the tasks for specific DSLs for our experiments. The five generalization tasks are:

1. ***Length-Generalization***: Can a model *produce longer code* than seen in training, when necessary? Here, "length" counts the number of subprograms and not the number of tokens, so there is more

---

[1] Ideally, operations within a group should have meaningful commonalities that form one concept, and each concept should have roughly equal semantic complexity, but these are not strictly required.

Figure 1: Our five compositional generalization tasks. Circles represent subprograms that join to form programs as train or test examples, colored circles represent subprograms of a particular concept or operation, and bold outlines represent analogous functionality of different operations.

    emphasis on generalizing to more complex compositional patterns. For this task, we train on problems of lengths 1 to $n$ and test on lengths $n+1$ to $m$ (where $m > n$).

2. ***Compose-Different-Concepts***: Can a model *use concepts in different combinations* than seen in training? Specifically, train the model on compositions of operations from the same concept, and test on compositions from different concepts. For example, if two concepts consist of operations $\{A_1, A_2, \ldots\}$ and $\{B_1, B_2, \ldots\}$, then the training programs have the form $A_i \circ A_j$ and $B_i \circ B_j$, and the testing programs have the form $A_i \circ B_j$ and $B_i \circ A_j$ (and similarly for compositions of 3 or more operations). A real-world example might be training on program containing only TensorFlow or only NumPy, but synthesizing code at test time using both libraries.

3. ***Switch-Concept-Order***: Can a model *compose concepts in different orders* than seen in training? We train on compositions of operations drawn from one sequence of concepts and test on a different sequence of concepts, e.g., train on $A_i \circ B_j$ and test on $B_i \circ A_j$. As a real-world example, in the training data a function might be validating inputs at the beginning of the code, but we want to use the function in a different context, e.g., to validate results at the end.

4. ***Compose-New-Operation***: Can a model learn to *use a new isolated operation within a larger composition*? In this task, we train on the isolated operation and compositions without the operation, and test on compositions using the operation. A real-world example of this kind of generalization would be composing a new function with others in a larger solution, after seeing examples of the function used in isolation.

5. ***Add-Operation-Functionality***: Can a model *extend its understanding of an operation by drawing on parallels* to other operations? We omit from the training data some functionality of an operation that could be inferred from other context, and test on programs using that functionality. This task can occur when a library function is upgraded with a new parameter whose behavior can be inferred from analogous parameters in other functions.

These five tasks can be grouped into three themes: (a) *length generalization*; (b) *mix and match concepts* (tasks 2 and 3): compose concepts in ways that were not seen during training; and (c) *apply general principles* (tasks 4 and 5): adapt to new, updated, or custom APIs.

## 3   Benchmark Creation

While Section 2 focused on the meta-benchmark describing five compositional generalization tasks, this section describes our instantiation of those tasks into compositional generalization datasets for two popular synthesis domains, RobustFill (Devlin et al., 2017) and DeepCoder (Balog et al., 2017).

**RobustFill.** In the RobustFill domain, the objective is to synthesize a sequence of string manipulation operations from I/O examples, where each example's input is a single string. A RobustFill program is a concatenation of expressions. There are 4 categories of expressions: operations that extract a substring from the input (e.g., `GetToken(`*`regex, index`*`)`), operations that return a modified version of the input (e.g., `ToCase(`*`case`*`)`), a special `Compose` operation (applying a modification operation to the result of another operation), or a constant string character. For example, the program `GetFrom(' ') | Const('.') | Compose(ToCase(PROPER), GetToken(WORD, 1))` is a

---

**Algorithm 1** ExeDec: synthesis via decomposition in the execution space.

Note, $\{x_i\}$ is short for $[x_1, \ldots, x_n]$ throughout, where $n$ is the number of I/O examples.

1:  **function** EXEDEC($\{(I_i, O_i)\}$)
2:      $t \leftarrow 1$
3:      $(I_i^{(1)}, O_i^{(1)}) \leftarrow (I_i, O_i),\ \forall i$
4:      **while** True **do**
5:          $\{S_i^{(t)}\} \leftarrow$ SUBGOALMODEL($\{(I_i^{(t)}, O_i^{(t)})\}$)          ▷ Predict the next execution subgoals
6:          $P^{(t)} \leftarrow$ SYNTHESIZERMODEL($\{(I_i^{(t)}, S_i^{(t)})\}$)          ▷ Predict the next subprogram
7:          $E_i^{(t)} \leftarrow$ EXECUTE($P^{(t)}, I_i^{(t)}$),$\ \forall i$
8:          **if** $\forall i.\ E_i^{(t)} = O_i^{(t)}$ **then**          ▷ Is this the last subprogram?
9:              **return** COMBINEPROGRAMPARTS($P^{(1)}, \ldots, P^{(t)}$)
            ▷ Update $\{(I_i^{(t)}, O_i^{(t)})\}$ to represent work that is left to be done (domain-specific).
10:         $(I_i^{(t+1)}, O_i^{(t+1)}) \leftarrow$ UPDATESPECIFICATION($I_i^{(t)}, O_i^{(t)}, E_i^{(t)}$),$\ \forall i$
11:         $t \leftarrow t + 1$

---

concatenation of 3 expressions and transforms the input string "TURING, Alan" into the output string "Alan.Turing". See Appendix A for the full RobustFill DSL, which we extended from the original RobustFill paper (Devlin et al., 2017) by adding more operations. Appendix B contains further details about our constructed datasets, including the different compositional generalization splits and the process for generating synthetic programming tasks according to those splits.

**DeepCoder.**    The DeepCoder domain involves manipulation of integer lists in a line-by-line programming style. Tasks have one or more inputs which may be integers or integer lists. Each line of a DeepCoder program applies one DSL operation to inputs or previous variables and assigns the result to a new variable. The result of the last line is the program's output. Operations include first-order list operations (`Sort`, `Reverse`, and various forms of indexing, slicing, and aggregating) and higher-order operations (Haskell-inspired `Map`, `Filter`, `Count`, `ZipWith`, and `Scan1l`) which manipulate lists using one of several hardcoded lambda functions. As an example, the program `x0 = INPUT | x1 = Map (**2) x0 | x2 = Sort x1` (where "|" denotes a new line) transforms the input list $[5, 3, -4]$ into the output list $[9, 16, 25]$. See Appendix A for the full DeepCoder DSL and Appendix B for more details about our instantiation in the DeepCoder domain.

**Choice of Domains.**    Both domains allow us to generate a large amount of synthetic training data with ground-truth decompositions into subprograms. For more realistic code in general-purpose programming languages, such data collection requires more effort, especially if "natural" decompositions are desired. Beyond the difference in string versus list manipulation, RobustFill and DeepCoder are quite different in other important ways, allowing us to study the compositional generalization of various approaches in different scenarios. First, RobustFill gradually builds an output by combining results of subprograms that are mostly independent, while DeepCoder applies operations repeatedly to the same few objects until the output is reached. In this sense, RobustFill is closer to inverse CAD (Ellis et al., 2019), instantiating complex objects with many fields like dataclasses, or other tasks involving several independent analyses, while DeepCoder is closer to tensor manipulation (Shi et al., 2022a), dynamic programming, or other tasks involving sequences of manipulations or updates applied to the same objects. Second, RobustFill uses the same input for each subprogram while DeepCoder involves program states that change due to the new variable bindings on each line, making DeepCoder more complex and closer to realistic programs with execution states changing over time.

## 4    PROGRAM SYNTHESIS VIA DECOMPOSITION

In this section we describe our proposed program synthesis method based on execution decomposition, where the model predicts step-by-step execution subgoals and synthesizes subprograms for each step.

**Execution Decomposition (ExeDec).**    The ExeDec strategy outlined in Algorithm 1 aims to reason about the step-by-step execution behavior of a program rather than the code tokens. As in Section 2, we assume that the program is a sequence of one or more *subprograms* that may be combined later. At each step, to synthesize the next subprogram, we first call a *SubgoalModel* that takes I/O

examples and predicts the next execution subgoals, i.e., the output of the next subprogram for each example. Because the subgoal is the desired output at this step, predicting the next subprogram is itself a PBE task. Thus, we provide the inputs and subgoals to a *SynthesizerModel* which predicts the corresponding subprogram. Finally, we execute the predicted subprogram and compute an *updated specification* that describes the work that remains to be done by the rest of the program.

This updated specification is maintained throughout the step-by-step synthesis process. Because the overall program is specified by I/O examples, we use I/O examples for the updated specification as well. Intuitively, the inputs in the updated specification will be the current program state, and the outputs will be the output of the overall task, but the details are slightly different because of specifics of the DSLs. We begin with the original I/O examples for the overall synthesis task, and we update them in a domain-specific way as subprograms are synthesized (line 10). For instance, in RobustFill the input for each subprogram is the same as the original input, while the output becomes smaller as we remove already-synthesized prefixes of the output: $(I_i^{(t+1)}, O_i^{(t+1)}) \leftarrow (I_i^{(t)}, \textsc{RemovePrefix}(O_i^{(t)}, E_i^{(t)}))$; this is because the top level operation in RobustFill programs is always concatenation.[2] For DeepCoder, the input is the full program state (i.e., the set of variables and their values for each example) which is expanded with new variables as subprograms are synthesized, while the output remains constant for each example: $(I_i^{(t+1)}, O_i^{(t+1)}) \leftarrow (I_i^{(t)} \cup E_i^{(t)}, O_i^{(t)})$. If ExeDec synthesizes a subprogram that executes to the entire remaining output, there are no more subprograms to synthesize, so the subprograms are combined to form the full synthesized program.

Algorithm 1 describes a single synthesis attempt, but we actually perform a *search* comprising multiple synthesis attempts running efficiently in parallel using a modified beam search where each beam state is a partial rollout of the step-by-step synthesis algorithm. Appendix C has more details.

**Model Architecture.** Recall from Algorithm 1 that ExeDec relies on two models, the SubgoalModel and SynthesizerModel. We let both be sequence-to-sequence (seq2seq) models, which have been shown to be successful on various natural language (Bahdanau et al., 2016; Vaswani et al., 2017) and program synthesis tasks (Devlin et al., 2017). We choose our seq2seq model to be a Transformer due to its impressive performance on natural language tasks over traditional RNNs (Vaswani et al., 2017). We modify the baseline Transformer architecture to account for the fact that we operate on sets of inputs due to having multiple I/O examples. We call our model a Specification-Transformer.

For consistent notation for the two models, we let $\{X_i\}$ be the multi-example input to the transformer and $Y$ its output. Formally, $X_i = (I_i, O_i)$ for SubgoalModel and $(I_i, S_i)$ for SynthesizerModel, and $Y = [S_1, \text{Sep}, S_2, \text{Sep}, \dots, S_n]$ for SubgoalModel and $Y = P$ for SynthesizerModel, where $\text{Sep}$ is a new token added to our vocabulary to partition the subgoals across examples. Note that subgoals $S_i$ and subprogram $P$ are sequences of tokens.

Our Specification-Transformer consists of two modules. A Transformer encoder receives the specification $\{X_i\}$ and produces an encoding $\phi$. Following Devlin et al. (2017), our encoder performs *double attention* on the specification. That is, for each example $X_i$, the encoder performs the operation $\phi_i \leftarrow \text{TransformerEncoder}(X_i)$, where the encoder performs self-attention on input $I_i$ followed by cross-attention from the output (either $O_i$ or $S_i$) to $I_i$. Then, the encoding $\phi$ is simply the concatenation across examples $\phi \leftarrow \text{Concat}(\{\phi_i\})$. Next, a Transformer decoder takes the encoding and autoregressively generates the output token-by-token. Formally, let $Y_{\ell-1} = [y_1, y_2, \dots, y_{\ell-1}]$ be the output (subgoals or subprogram) generated so far. The decoder predicts the next output token as $y_\ell \leftarrow \text{TransformerDecoder}(Y_{\ell-1}, \phi)$. As described by Vaswani et al. (2017), the Transformer encoder and decoder both apply a stack of self-attention and feed-forward units. For the SubgoalModel, we use Aligned Relative Attention (ARA), a new technique that helps the model output a *sequence of sequences* (a subgoal for each I/O example, concatenated together); see Appendix D for details.

**No-Subgoal Ablation.** We also experiment with an ablation of ExeDec that performs step-by-step decomposition but without predicting execution subgoals first, instead directly predicting the next subprogram from the I/O examples. In Algorithm 1, this ablation is achieved by replacing lines 5 and 6 with a single line, $P^{(t)} \leftarrow \textsc{CombinedModel}(\{(I_i^{(t)}, O_i^{(t)})\})$, thus skipping the step of predicting execution subgoals. This ablation uses the same model architecture (without ARA) and an analogous

---

[2]If the synthesized subprogram does not execute to a prefix of the current output for all examples, this synthesis attempt cannot succeed due to RobustFill's concatenation of subprograms. Such "invalid" subprograms are detected and handled during a beam search.

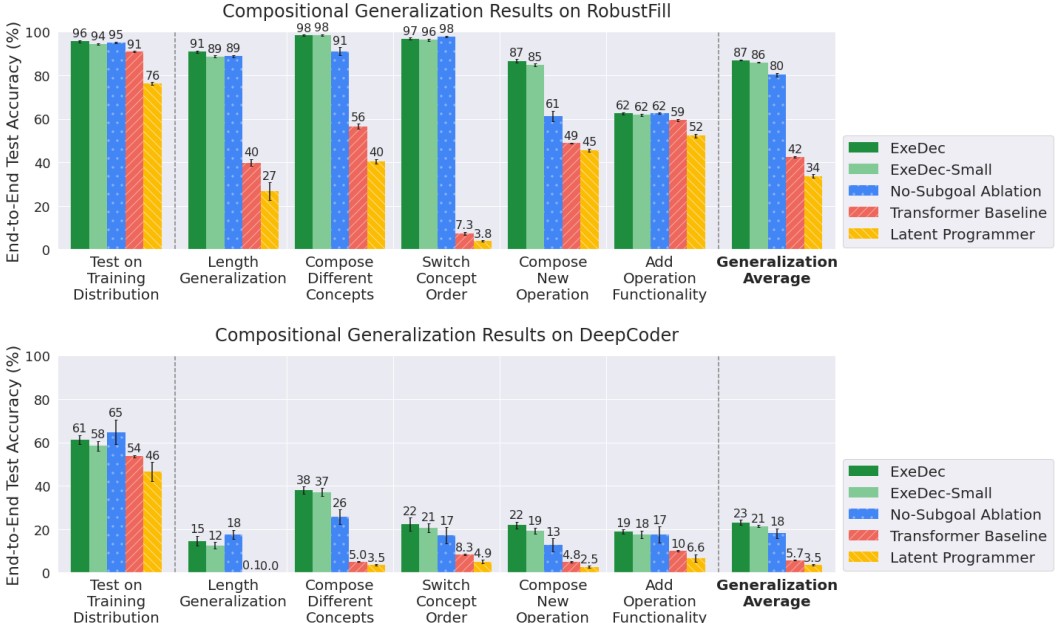

Figure 2: Compositional generalization results with beam size 10. Error bars denote 95% confidence intervals of the mean across 5 trials. On both datasets, ExeDec generalizes better than the no-subgoal ablation, while both decomposition variations greatly outperform the Transformer baseline.

beam search. Several prior works (Zohar & Wolf, 2018; Ellis et al., 2019; Chen et al., 2019) perform synthesis step-by-step, providing execution feedback to the synthesizer after each step to inform future predictions. Our ablation captures the essence of those approaches adapted to our setting.

**Model Training.** We generate training problems as described in Section 3. We train the ExeDec and ablation models using *decomposed* data, that is, based on teacher forcing using Algorithm 1. Specifically, for each subprogram in the ground-truth solution, we collect (A) the updated specification based on executing the previous ground-truth subprograms, (B) the subprogram's execution result on all examples, and (C) the subprogram itself. Then, we train the SubgoalModel to predict (B) given (A), the SynthesizerModel to predict (C) given (B) and the example inputs from (A), and the CombinedModel to predict (C) given (A). Each model type is trained separately for each generalization task. Appendix E contains more training details, including model sizes and hyperparameters.

# 5 EXPERIMENTS

We experiment with Transformers trained from scratch and with LLMs using few-shot prompting.

## 5.1 TRANSFORMERS TRAINED FROM SCRATCH

These experiments compare ExeDec, a version with smaller models called ExeDec-Small, the no-subgoal ablation, a Transformer baseline without any decomposition, and Latent Programmer (Hong et al., 2021). All models use the same hyperparameters and architecture except: (1) ExeDec-Small and Latent Programmer use smaller models (details and reasoning in Appendix E), (2) ARA only applies to the SubgoalModel, and (3) because the baseline Transformer and Latent Programmer are trained on entire programs instead of subprograms, but the number of training examples is held constant, they actually see more subprograms during training than our models.

Using our compositional generalization datasets (Section 3) and models (Section 4), we ran the different approaches and measured their overall success rate on 1000 test examples per generalization task. We repeated the experiments using 5 different random initializations for model training. Figure 2 shows the results when using a beam size of 10, Appendix F contains results with beam size 1, and Appendix G analyzes the accuracy of individual steps.

**Discussion.** On both domains, ExeDec significantly outperforms the Transformer baseline on every generalization task and in the i.i.d. setting (testing on the training distribution without any compositional generalization). Specifically, ExeDec achieves $+44\%$ higher average compositional generalization than the Transformer baseline on RobustFill and $+18\%$ on DeepCoder, a *4.4× higher* success rate. But despite the notable improvements, DeepCoder in particular remains a difficult domain with deeply nested operation compositions that obscure the intended computation, while RobustFill has a more flat compositional structure that is easier to learn.

Our step-by-step decomposition approach introduces important inductive biases into the approach. By training models on the decomposed data, we teach the models that subprograms can be reasoned about separately, regardless of the compositional patterns present in other subprograms. The SubgoalModel does not see any code tokens and is only affected by compositional generalization patterns indirectly (since the distribution over programs affects the distribution over execution traces), and the SynthesizerModel only sees code tokens for the current subprogram and cannot reference any compositional patterns that appear when comparing to other subprograms. In contrast, the Transformer baseline sees all compositional patterns in the full programs, making it more likely to overfit to those patterns. The decomposition strategy also encourages our models to understand intermediate program states while the Transformer baseline is not trained with such execution information.

Compared to the no-subgoal ablation, ExeDec achieves higher compositional generalization performance on a majority of generalization tasks across the two domains, averaging $+7\%$ improvement on RobustFill (a 34% reduction in failures) and $+5\%$ on DeepCoder (a $1.28\times$ multiplicative improvement). This supports our hypothesis that predicting execution states is more robust than predicting code in the compositional generalization setting. ExeDec-Small performs slightly worse than ExeDec (1.4% worse on average and up to 3% worse on any individual generalization task) but ExeDec-Small still significantly outperforms the other approaches overall.

Even though ExeDec performs the best in most situations, the no-subgoal variation is slightly better on DeepCoder's training distribution and *Length-Generalization*. Appendix H provides some intuition on "spurious patterns" related to this result. In theory, one could combine the two decomposition variations in an ensemble to get the best of both approaches on unknown test distributions. Finally, we observe that in most cases ExeDec has smaller variance across random initializations than the no-subgoal variation, i.e., ExeDec might be more consistent in practice.

As a case study, we compare ExeDec, the no-subgoal ablation, and the Transformer baseline on example RobustFill and DeepCoder problems in Appendix I. Through these examples, we discuss some behaviors and observations that clarify the advantages to ExeDec's approach.

## 5.2 LLM EXPERIMENTS

It is fundamentally difficult to measure compositional generalization in LLMs, because compositional generalization is a function of the relationship between the training and test distributions, but in LLMs it is not easy to control the pretraining data. However, we have more control in a few-shot prompting setup, as long as we focus on program concepts that cannot have occurred in the pretraining data set. Based on this insight, in these experiments, we used our benchmarks to measure the compositional generalization ability of PaLM 2 Unicorn (Google et al., 2023) during few-shot prompting for PBE. We use the same compositional generalization splits for DeepCoder and RobustFill, except that the few-shot examples and test problems have length at most 3. We make the problems easier because LLMs in general perform poorly on program synthesis tasks specified only through I/O examples, compared to natural language specifications. Within each split we balance the distribution of program lengths as much as possible,[3] and we use 200 test problems per generalization task. Each prompt contains a description of the DSL including the available functionality, followed by 4 few-shot examples of PBE tasks and solutions drawn from the training split (different tasks are randomly chosen for different test problems), followed by the specification for a test problem (see Appendix J).

To make the tasks better suited to LLMs, we transform our DSL programs into Python functions that call a hypothetical `dsl` library to access the DSL functionality. The RobustFill subprogram `GetToken(WORD, 1)` becomes `dsl.GetToken(x, dsl.Type.Word, 1)`, and the DeepCoder

---

[3]For example, *Compose-Different-Concepts*, *Switch-Concept-Order*, and *Compose-New-Operation* all require programs of length at least 2, so these tasks have a 50/50 split between programs of lengths 2 and 3.

Table 1: Compositional generalization results for the LLM experiments. Each cell contains the number of solved tasks out of 200 test problems. For approaches, @1 means 1 greedy decoding and @5 means using 5 samples with temperature 0.4. For columns, "None" means no generalization, "Gen. Tasks" refers to the 5 compositional generalization tasks in the order given in Section 2 (consistent with the other figures), and "Avg" is the average across the 5 generalization tasks.

| Approach | RobustFill None | Gen. Tasks | | | | | Avg | DeepCoder None | Gen. Tasks | | | | | Avg | DeepCoder-Pythonic None | Gen. Tasks | | | | | Avg |
|---|---|---|---|---|---|---|---|---|---|---|---|---|---|---|---|---|---|---|---|---|---|
| Baseline @1 | 4 | 4 | **0** | 1 | 0 | 0 | 1.0 | 23 | 0 | 1 | 6 | 0 | 5 | 2.4 | 25 | 1 | 0 | 11 | 0 | 4 | 3.2 |
| Ablation @1 | 16 | 4 | **0** | 0 | 1 | **6** | 2.2 | 31 | 1 | 2 | **13** | 3 | 5 | 4.8 | 30 | **4** | 0 | 12 | 4 | 4 | 4.8 |
| ExeDec @1 | **21** | **5** | 0 | 0 | 1 | **6** | **2.4** | **36** | **2** | **3** | 11 | **4** | **10** | **6.0** | **46** | 3 | **3** | **15** | **5** | **16** | **8.4** |
| Baseline @5 | 15 | 1 | 0 | **1** | 2 | 5 | 1.8 | 36 | 0 | 1 | 11 | 5 | 13 | 6.0 | 34 | 2 | 1 | 15 | 5 | 8 | 6.2 |
| Ablation @5 | 29 | **7** | **1** | 0 | 4 | 7 | 3.8 | 51 | 1 | 4 | **18** | 8 | **21** | **10.4** | 42 | 4 | 8 | 19 | 10 | 11 | 10.4 |
| ExeDec @5 | **32** | 5 | 0 | 0 | **5** | **10** | **4.0** | **56** | **2** | **5** | 17 | 8 | 17 | 9.8 | **59** | **5** | **9** | **25** | **12** | **30** | **16.2** |

subprogram `x2 = Map (**2) x1` becomes `x2 = dsl.Map(dsl.SQUARE, x1)`. For DeepCoder, we alternatively try using Pythonic expressions for all DSL functionality except the `Scanl1` operation, which is difficult to inline; the previous example then becomes `x2 = [x ** 2 for x in x1]`.

By representing DSL programs as Python functions in this way, we enable the LLM to draw upon its general understanding of Python from its pretraining data, while requiring the LLM to use a new Python library from only a description of the library along with 4 few-shot examples. This setting mirrors realistic use-cases where a user asks about a new, custom, or proprietary library that the LLM was not trained on. Appendix J contains examples of our prompts and Python-style programs. The LLM is allowed to use arbitrary Python, although it usually follows the style in the examples.

We experimented with three prompting approaches analogous to the other experiments:

1. The baseline approach is to predict the entire solution program in one decoding.
2. The Ablation-style approach predicts the program step-by-step. Given the problem specification and history of previous steps, the LLM predicts the next line of code. We then execute the program-so-far and concatenate the predicted line of code along with its execution results into the history portion of the prompt, which will influence future steps. This stepwise process continues until the desired outputs are reached, the program fails to execute, or a budget of 3 steps is exhausted.
3. The ExeDec-style approach is similar, except that at each step, the LLM predicts the *next execution subgoal* followed by a line of code for that step (analogous to calling the SubgoalModel and SynthesizerModel). Note that the LLM's subgoal prediction might be inconsistent with the predicted code, so in the history of previous steps, we replace the predicted subgoals with the actual execution results (analogous to how the specification is updated in ExeDec). Over multiple steps, this process creates a prompt almost identical to that of the Ablation-style approach, except that ExeDec-style has the execution results of a step *before* the code for that step, while the Ablation-style has the execution results *after* the code.

The results are in Table 1. The ExeDec-style prompting strategy leads to the best performance for all no-generalization cases, and all but one case for the generalization average. Also, the ExeDec-style approach significantly improves when programs are written in a more natural form (going from DeepCoder to DeepCoder-Pythonic), which is a promising sign for its general applicability. For DeepCoder-Pythonic, the ExeDec-style approach solves between 40% and 75% more tasks than the next-best approach, considering each combination of no-generalization vs. generalization average and greedy decoding vs. pass@5 sampling. But despite these improvements, compositional generalization remains difficult for LLMs. Appendix K discusses common failure modes in the LLM experiments.

## 6 RELATED WORK

**Compositional Generalization.** Compositional generalization is well-studied in NLP, with established benchmarks evaluating the understanding of natural language sentences with compositionally novel structures, either constructed by synthesizing examples based on predefined generalization patterns similar to this work (Lake & Baroni, 2018; Bahdanau et al., 2019), or by partitioning *i.i.d.* samples into splits with disjoint compositional structures (Finegan-Dollak et al., 2018; Keysers

et al., 2020; Shaw et al., 2021). Our benchmark takes inspiration from SCAN (Lake & Baroni, 2018) and COGS (Kim & Linzen, 2020), which define a taxonomy of compositional patterns in natural language. While some generalization concepts are similar to those in Section 2, we focus on measuring compositional generalization of computer programs using I/O examples without natural language utterances, whose compositional structures are quite different from those in natural language.

To improve compositional generalization in natural language understanding, earlier works have proposed specialized task-dependent neural architectures (Russin et al., 2019; Li et al., 2019; Liu et al., 2020; Chen et al., 2020; Herzig & Berant, 2020). More generalized approaches include meta-learning (Lake, 2019; Wang et al., 2021a; Conklin et al., 2021) and data augmentation (Andreas, 2020; Oren et al., 2021; Akyürek et al., 2021; Wang et al., 2021b; Qiu et al., 2022). There have also been recent attempts in improving the compositional generalization capabilities of large language models via representation learning (Furrer et al., 2020; Herzig et al., 2021) and in-context learning (Zhou et al., 2023; Drozdov et al., 2023).

In machine learning for code, some works include length generalization results (Bieber et al., 2020; Balog et al., 2017; Ellis et al., 2019), and Nye et al. (2021) use compositional generalization in some experiments, but we study compositional generalization in a much more systematic manner.

**Programming by Example.** Various techniques have been applied to program synthesis (Gulwani et al., 2017), and recently much attention has focused on machine learning for programming by example (Devlin et al., 2017; Parisotto et al., 2017; Ellis et al., 2021). Many methods incorporate learning to guide the search over programs, such as using learned premise selection (Balog et al., 2017; Odena & Sutton, 2020), syntax-guided search (Yin & Neubig, 2017; Lee et al., 2018), bottom-up search (Shi et al., 2022a; Barke et al., 2020), two-level search (Nye et al., 2019), and execution-guided synthesis methods (Odena et al., 2020; Shi et al., 2022b).

**Multi-step Program Synthesis.** ExeDec is an instance of multi-step program synthesis, which broadly refers to methods involving multiple calls to (potentially different) models. *Execution-guided synthesis* is a popular form of this, iteratively generating and refining partial programs using execution information (Zohar & Wolf, 2018; Ellis et al., 2019; Chen et al., 2019; Shrivastava et al., 2021), and some approaches do this with latent representations of the program state (Chen et al., 2021b) or execution traces (Shin et al., 2018). *Planning* is another form of multi-step synthesis that first generates high-level plans of what the program should do (Nye et al., 2019; Murali et al., 2018; Zhang et al., 2023), sometimes with latent representations of plans (Hong et al., 2021). Our method, ExeDec, draws ideas from both avenues of multi-step synthesis, making plans by predicting subgoals and using step-by-step program execution to guide the search.

## 7 CONCLUSION

We explored the important aspect of compositional generalization in neural program synthesis. The ability to decompose complex tasks into smaller subtasks is a fundamental skill employed by human programmers, and measuring whether neural program synthesis methods exhibit similar capabilities is crucial for assessing their potential. We introduced a meta-benchmark that characterizes 5 forms of compositional generalization in program synthesis, and we instantiated these generalization tasks in the RobustFill and DeepCoder domains. The findings demonstrate that the ExeDec approach of predicting decompositions of program execution, rather than solely focusing on program syntax, leads to significantly improved compositional generalization for both Transformers trained from scratch and LLMs in a few-shot setting. This suggests that incorporating information about the step-by-step decomposition and leveraging it in the synthesis of programs can enhance the ability of neural models to tackle more complex tasks. Even so, compositional generalization remains challenging for neural program synthesizers, and our meta-benchmark can help measure continued progress in this area.

**Limitations.** One limitation of ExeDec is its need for a training dataset with ground-truth decompositions. Our experiments used synthetic programs with line-by-line decomposition, but perhaps better results could be obtained with a dataset containing more *natural* decompositions. Furthermore, the line-by-line decomposition could be a limitation as programmers often think in larger chunks or hierarchically; Appendix L discusses a potential hierarchical formulation of ExeDec to address this limitation in future work. Lastly, our SubgoalModel predicts tokenizations of objects, but to handle more complex objects, a more general SubgoalModel might instead predict *abstractions* of objects.

## REPRODUCIBILITY STATEMENT

Our code, datasets, and checkpoints for the Transformer models trained from scratch are available at https://github.com/google-deepmind/exedec. Additionally, Appendix E contains details about model hyperparameters and sizes for the models we trained.

## ACKNOWLEDGEMENTS

The authors would like to thank Xinyun Chen, Martin Abadi, Rif Saurous, and the anonymous reviewers for their helpful comments.

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

# Appendices

## A ROBUSTFILL AND DEEPCODER DSLS

Figure 3 contains the DSL for the RobustFill experiments. See Devlin et al. (2017) for a description of what the operations do. We add the operations `Substitute`, `SubstituteAll`, which replace the $i^{\text{th}}$ occurrence (or all occurrences) of a regex $r$ with character $c$, and `Remove`, `RemoveAll`, which remove the $i^{\text{th}}$ occurrence (or all occurrences) of a regex $r$, to increase the expressivity of our DSL.

Figure 4 shows the DSL used for the DeepCoder experiments. The operations are exactly as described in Balog et al. (2017).

$$
\begin{array}{rcl}
\text{Program } P & := & \texttt{Concat}(e_1, e_2, \ldots) \\
\text{Expression } e & := & s \mid m \mid o \mid \texttt{ConstStr}(c) \\
\text{Compose } o & := & m_1(m_2) \mid m(s) \\
\text{Substring } s & := & \texttt{SubStr}(k_1, k_2) \mid \texttt{GetSpan}(r_1, i_1, b_1, r_2, i_2, b_2) \mid \texttt{GetToken}(r, i) \\
& & \mid \texttt{GetUpto}(r) \mid \texttt{GetFrom}(r) \\
\text{Modification } m & := & \texttt{ToCase}(a) \mid \texttt{Replace}(\delta_1, \delta_2) \mid \texttt{Trim}() \mid \texttt{GetFirst}(r, i) \mid \texttt{GetAll}(r) \\
& & \mid \texttt{Substitute}(r, i, c) \mid \texttt{SubstituteAll}(r, c) \mid \texttt{Remove}(r, i) \mid \texttt{RemoveAll}(r) \\
\text{Regex } r & := & \texttt{NUMBER} \mid \texttt{WORD} \mid \texttt{ALPHANUM} \mid \texttt{ALL\_CAPS} \mid \texttt{PROPER\_CASE} \mid \texttt{LOWER} \mid \texttt{DIGIT} \mid \texttt{CHAR} \mid \delta \\
\text{Case } a & := & \texttt{ALL\_CAPS} \mid \texttt{PROPER\_CASE} \mid \texttt{LOWER} \\
\text{Position } k & := & -100 \mid -99 \mid \ldots \mid -1 \mid 0 \mid 1 \mid 2 \mid \ldots \mid 100 \\
\text{Index } i & := & -5 \mid -4 \mid \ldots \mid -1 \mid 1 \mid 2 \mid \ldots \mid 5 \\
\text{Boundary } b & := & \texttt{START} \mid \texttt{END} \\
\text{Character } c & := & A \mid \ldots \mid Z \mid a \mid \ldots \mid z \mid 0 \mid \ldots \mid 9 \mid \delta \\
\text{Delimiter } \delta & := & \texttt{\&,.?!@()[]\%\{\}/:;\$\#\ "'}
\end{array}
$$

Figure 3: The DSL for string transformation tasks in the RobustFill domain, slightly modified from Devlin et al. (2017) to add more functionality.

$$
\begin{array}{rcl}
\text{Program } P & := & i_1; i_2; \ldots; a_1; a_2; \ldots \\
\text{Initialization } i & := & v \leftarrow \texttt{INPUT} \\
\text{Assignment } a & := & v \leftarrow f \mid v \leftarrow h \\
\text{First-Order Operation } f & := & \texttt{Head}(l) \mid \texttt{Last}(l) \mid \texttt{Access}(n, l) \mid \texttt{Minimum}(l) \mid \texttt{Maximum}(l) \mid \texttt{Sum}(l) \\
& & \mid \texttt{Take}(n, l) \mid \texttt{Drop}(n, l) \mid \texttt{Reverse}(l) \mid \texttt{Sort}(l) \\
\text{Higher-Order Operation } h & := & \texttt{Map}(\lambda, l) \mid \texttt{Filter}(\beta, l) \mid \texttt{Count}(\beta, l) \mid \texttt{ZipWith}(\Sigma, l, l) \mid \texttt{Scan1}(\Sigma, l) \\
\text{int} \to \text{int Lambda } \lambda & := & (+1) \mid (-1) \mid (*2) \mid (/2) \mid (*(-1)) \mid (**2) \mid (*3) \mid (/3) \mid (*4) \mid (/4) \\
\text{int} \to \text{bool Lambda } \beta & := & (>0) \mid (<0) \mid (\%2 == 0) \mid (\%2 == 1) \\
(\text{int, int}) \to \text{int Lambda } \Sigma & := & (+) \mid (-) \mid (*) \mid (\min) \mid (\max) \\
\text{Integer Variable } n & := & v \\
\text{List Variable } l & := & v \\
\text{Variable Name } v & := & x_1 \mid x_2 \mid \ldots
\end{array}
$$

Figure 4: The DSL for integer and list manipulation tasks in the DeepCoder domain, originally proposed in Balog et al. (2017).

# B   ROBUSTFILL AND DEEPCODER BENCHMARK DETAILS

**RobustFill.**   To create a task for our RobustFill datasets, we first sample random input strings up to 20 characters, one string for each of 4 examples. We then sample a program according to the train or test distribution for the generalization task (as described below), such that the program executes successfully on the inputs to form the example outputs. Due to the concatenation structure of RobustFill programs, we treat each concatenated expression as a subprogram, and recall that we denote the *length* of a program to be the number of subprograms.

For *Length-Generalization*, we train on programs of length 1 to 6 inclusive and test on programs of length 7 to 10. For *Compose-Different-Concepts*, we group together all of the substring operations into a *substring concept* and all of the modification operations plus constant strings as a *non-substring concept* (the `Compose` operation is omitted), using programs of length 2-6 for both train and test. We use the same lengths and concepts for *Switch-Concept-Order*, where training tasks use only the substring concept for the first half of the parts and only the non-substring concept for the latter half, and test tasks have the ordering reversed. For *Compose-New-Operation*, 25% of training tasks are length 1 programs containing only a `Compose` operation, the remainder of the training tasks are length 2-6 programs without `Compose`, and we test on length 2-6 programs that use `Compose`. For *Add-Operation-Functionality*, all tasks are length 1-6 programs, we train on those where a substring operation is *not* used within a `Compose` operation, and we test on programs where a substring operation *is* used within a `Compose` operation.

**DeepCoder.**   For DeepCoder, we treat each non-input line as a subprogram, so the example program above has length 2. We use 3 I/O examples, at most 2 inputs, lists with at most 5 elements, and integers between $-50$ and $50$ inclusive. For *Length-Generalization*, we train on programs of length 1 to 4 and test on length 5. For *Compose-Different-Concepts* and *Switch-Concept-Order*, we use programs of length 1 to 4 and split the operations into a concept containing all first-order operations plus the `Map` operation, and another concept containing all remaining higher-order operations. For *Compose-New-Operation*, 25% of training tasks are length 1 programs containing only a `Scanl1` operation, the remainder of training tasks are length 2-4 programs without `Scanl1`, and we test on length 2-4 programs that use `Scanl1`. For *Add-Operation-Functionality*, all tasks are length 1-4 programs, `Scanl1` is only used with the lambdas `(-)` and `(min)` during training, and we test on tasks where `Scanl1` is used with the other lambdas `(+)`, `(*)`, and `(max)`.

The program sampling procedure that we used for RobustFill ensures that the ground-truth program for a test task is within the test distribution over programs. However, the task might actually have a different solution within the *train* distribution. Then, solving this test task is not necessarily a signal of generalization.[4] To address this, we construct the DeepCoder dataset more carefully as follows. We sample random inputs as before. Then, we perform an exhaustive enumerative search of all programs in the train distribution up to a maximum length, and similarly for programs in the test distribution. As a result, we can identify all minimal-length solution programs for a given task (if it is solvable by any program up to the maximum enumerated length). Finally, we sample training tasks among those where there exists a minimal-length solution program in the train distribution, and test tasks among those where *all* minimal-length solutions are in the test distribution.[5]

# C   BEAM SEARCH DURING EXEDEC

Section 4 and Algorithm 1 describe a single synthesis attempt, but we design a beam search process to run multiple synthesis attempts efficiently in parallel.

Our goal is the same as in traditional beam search (Sutskever et al., 2014), which is to output $k$ sequences that maximize a score function. Formally, a candidate sequence $C$ at step $t$ is a sequence of execution subgoals and subprograms $C = [\{S_i^{(1)}\}, P^{(1)}, \ldots, \{S_i^{(t)}\}, P^{(t)}]$. As notation, we let $C^{(j)}$ denote the prefix of $C$ up to and including the $j$-th subprogram $P^{(j)}$. The score for a candidate

---

[4]This issue is far less prevalent in the RobustFill dataset due to the concatenation program structure reducing the number of different programs that solve a task.

[5]Note that it is still possible for a test task to be solved with a *longer* program in the train distribution, but detecting these cases in general is extremely difficult.

sequence $C$ is given by

$$\mathsf{Score}(C) = \sum_{j=1}^{t} \mathsf{Score}(\{S_i^{(j)}\}) + \mathsf{Score}(P^{(j)}) + \mathsf{Valid}(C^{(j)}) . \tag{1}$$

Here, since both SubgoalModel and SynthesizerModel in ExeDec are autoregressive sequence models (see Section 4 for more details), they can produce token sequences (either set of subgoals or a subprogram) with their corresponding log-probabilities which we use as the score function, i.e., $\mathsf{Score}(\{S_i^{(j)}\}) = \log p(S_i^{(j)} \mid \{(I_i^{(j)}, O_i^{(j)})\})$ and $\mathsf{Score}(P^{(j)}) = \log p(P^{(j)} \mid \{(I_i^{(j)}, S_i^{(j)})\})$. By default, $\mathsf{Valid}(C^{(j)}) = 0$ unless $C^{(j)}$ fails to meet some required conditions. First, we set $\mathsf{Valid}(C^{(j)}) = -\infty$ if any subgoal or subprogram in $C^{(j)}$ is a token sequence that does not parse, or if any subprogram does not execute successfully. We also check whether $C^{(j)}$ leads to unique program functionality within the beam of $k$ candidate sequences. Note that different candidate sequences can result in the same subprograms even if the subgoals are different, and furthermore, combining different subprograms might result in full programs with the same functionality that are practically equivalent for future steps. Therefore, we set $\mathsf{Valid}(C^{(j)}) = -\infty$ for all candidate sequences in the beam with corresponding program functionality equivalent to that of another higher-scoring candidate sequence in the beam. Finally, domain-specific checks may determine that the candidate sequence $C^{(j)}$ is unlikely to result in a successful program in later steps, or that some computation limit such as a maximum number of steps is reached. In any case, an "invalid" beam element will drop out of the beam at the next step, freeing space in the beam for expansions of other surviving beam elements.

Note that we do not perform separate beam searches for each call to a model, since an isolated beam search would start with a single empty beam element with score 0. Instead, we perform a single long beam search throughout Algorithm 1 where each beam element is a candidate sequence $C$ with score $\mathsf{Score}(C)$. Model calls can be seen as "extending" the ongoing beam search, adding more tokens to the existing candidate sequences while maintaining their scores.

## D  ALIGNED RELATIVE ATTENTION FOR THE SUBGOALMODEL

In the Specification-Transformer (Section 4), we use *relative attention* (Shaw et al., 2018), i.e., representations of relative positions (distances between positions) instead of absolute positions of tokens. This improves Transformer performance, particularly in length generalization (Shaw et al., 2018; Csordás et al., 2021). However, unlike the other models, our SubgoalModel actually predicts a sequence of sequences, i.e., a subgoal for each I/O example, concatenated together. Thus, the relative position between subgoal $S_i$ and I/O encoding $\phi_i$ is not consistent for every example index $i$, so a naive relative position embedding would not represent consistent information across examples. To remedy this, we propose *aligned relative attention* (ARA), which alters the "positions" of subgoal tokens in the relative distance calculation. Specifically, we increment the current position by 1 for each token during subgoal prediction as usual, but when starting a new subgoal $S_i$ we set the current position to the position of $\phi_i$. This ensures that the relative position between the beginnings of $S_i$ and $\phi_i$ is always 0. ARA only applies to the SubgoalModel to help it perform the sequence-of-sequences prediction, and it is not intended to address compositional generalization directly.

## E  TRAINING DETAILS

For the experiments involving Transformer models trained from scratch (Section 5.1), we performed a small hyperparameter search varying the learning rate and model size, choosing a single setting that performed well on training metrics across generalization tasks and model types. We used an embedding dimension of 512, hidden dimension of 1024, 3 layers, and 4 attention heads. For relative attention, we use 32 buckets for relative position embeddings, with bucket boundaries placed logarithmically given the maximum relative distance which is computed based on the lengths of the input and output sequences. We train with the Adam optimizer with a learning rate of $2 \times 10^{-4}$ with linear warmup for 16,000 steps and square root decay, with a batch size of 128 and 500K training steps, using fresh synthetic training data without repeating examples. Training took about 1 day for RobustFill (or about 5 hours for DeepCoder) with 8 TPU v2 accelerators per model.

In our experiments, some models have different sizes (with the other hyperparameters held constant):

- ExeDec, the no-subgoal ablation, and the Transformer baseline all use an embedding dimension of 512 and hidden dimension of 1024, as selected from the hyperparameter search mentioned above.

- ExeDec-Small uses an embedding dimension of 360 and hidden dimension of 720, such that the total number of parameters between the SubgoalModel and SynthesizerModel is approximately equal to the number of parameters in the no-subgoal ablation or in the baseline, which only use one model each. ExeDec's code solution only comes from the SynthesizerModel, while the SubgoalModel's predictions do not show up anywhere in the solution and only encourage the SynthesizerModel to think about the next subgoal instead of the end goal. Thus, compared to the ablation and baseline, ExeDec holds the SynthesizerModel size constant to compare the two modes of thinking, while ExeDec-Small holds the total number of parameters constant to verify that the improvements are not only due to extra parameters in the SubgoalModel.

- Latent Programmer uses a smaller embedding dimension of 256 and hidden dimension of 512 because we ran into out-of-memory issues when using the Latent Programmer's training script which trains two models at once. Our trained Latent Programmer models are still larger than those reported in the Latent Programmer paper leading to a corresponding performance increase, reaching 76% accuracy on no-generalization RobustFill compared to 57% reported in the Latent Programmer paper.

Additionally, for Latent Programmer, we performed a separate hyperparameter search varying the learning rate and number of pretraining steps. We chose a learning rate of $5 \times 10^{-4}$ for RobustFill and $2 \times 10^{-4}$ for DeepCoder, and 20,000 pretraining steps for both datasets. We use a beam size of 10 and a latent beam size of 3, as in the Latent Programmer paper. (For experiments with beam size of 1, the latent beam size is also 1.) All other settings were kept the same as for the other models.

For DeepCoder, the ground-truth training programs have variable names chosen randomly among a fixed set of possible names. If the variable names were used in a canonical order instead, the models would be unable to predict subprograms using variable names unseen during training, as is required for length generalization. When predicting a subprogram, the model only predicts the right hand side of the assignment, which at test time is assigned to a new variable name using a canonical ordering.

## F  BEAM SIZE 1 RESULTS FOR MODELS TRAINED FROM SCRATCH

Figure 2 in Section 5.1 shows detailed results for all generalization tasks, using beam size 10. We also performed experiments with beam size 1, summarized below in Table 2.

Table 2: Results for beam size 1 and beam size 10 (end-to-end test accuracy as percentages). "NoGen" refers to the case where no compositional generalization is required, while "GenAvg" refers to the average across the 5 compositional generalization tasks.

| | RobustFill | | | | DeepCoder | | | |
| | Beam Size 1 | | Beam Size 10 | | Beam Size 1 | | Beam Size 10 | |
| Approach | NoGen | GenAvg | NoGen | GenAvg | NoGen | GenAvg | NoGen | GenAvg |
|---|---|---|---|---|---|---|---|---|
| ExeDec | 81.4 | **73.1** | **95.5** | **87.0** | 39.1 | **8.4** | 61.2 | **23.1** |
| ExeDec-Small | 78.7 | 71.2 | 94.4 | 85.9 | 36.1 | 7.7 | 58.4 | 21.4 |
| Ablation | 79.5 | 63.8 | 94.9 | 80.2 | 39.9 | 5.6 | **64.7** | 18.1 |
| Transformer | **83.7** | 33.9 | 90.9 | 42.4 | **50.7** | 4.9 | 53.6 | 5.7 |
| Latent Programmer | 66.0 | 26.3 | 76.1 | 33.7 | 40.5 | 2.4 | 46.5 | 3.5 |

We observe that ExeDec achieves the highest generalization average, for both RobustFill and DeepCoder and for both beam size 1 and 10. Additionally, for both no-generalization and average generalization, the baseline Transformer's performance does not improve that much going from beam size 1 to 10. On the other hand, ExeDec improves significantly more from beam size 1 to 10, showing the effectiveness of our modified beam search (Appendix C) that enables exploring different solution routes by planning in the execution space.

## G  SINGLE-STEP ACCURACY

To gain more insight into the step-by-step synthesis process, we measured the "single-step accuracy" for ExeDec's SubgoalModel and SynthesizerModel, and the CombinedModel from the no-subgoal ablation. More specifically, for a single step in a trajectory that matches the ground-truth so far, how often does the SubgoalModel predict subgoals exactly matching the ground-truth ones (for all I/O examples) with a single greedy decoding? And, given a ground-truth trajectory so far, how often does the SynthesizerModel or CombinedModel predict a program whose behavior matches the ground-truth subprogram? The single-step accuracy results are in Table 3.

Table 3: Single-step accuracy percentages.

| Generalization Task | RobustFill | | | DeepCoder | | |
|---|---|---|---|---|---|---|
| | Subgoal | Synthesizer | Combined | Subgoal | Synthesizer | Combined |
| No Generalization | 97.7 | 97.1 | 94.9 | 55.8 | 98.9 | 64.4 |
| *Length-Generalization* | 97.2 | 96.4 | 94.0 | 31.6 | 96.4 | 41.9 |
| *Compose-Different-Concepts* | 95.3 | 98.9 | 90.5 | 38.0 | 94.3 | 40.0 |
| *Switch-Concept-Order* | 90.8 | 98.9 | 87.4 | 16.4 | 77.8 | 17.9 |
| *Compose-New-Operation* | 93.9 | 94.0 | 84.4 | 40.8 | 91.3 | 44.4 |
| *Add-Operation-Functionality* | 94.0 | 84.4 | 82.3 | 40.3 | 60.0 | 41.5 |

Note that the single-step accuracy metric is particularly low for the SubgoalModel and Combined-Model on DeepCoder because there are potentially many correct ways of solving the problem, and this metric only considers the single ground-truth solution. In fact, the SubgoalModel does not need to have super high accuracy in order for ExeDec to achieve good end-to-end results, because the SynthesizerModel can ignore slight errors in the subgoals and still produce a program that behaves as closely as possible while using only 1 DSL operation. We have seen many concrete cases of the SynthesizerModel being given a slightly incorrect subgoal and then producing the correct subprogram anyway, which disagrees with the subgoal but correctly makes progress overall.

## H  INTUITION OF SPURIOUS PATTERNS

In Figure 2, ExeDec sometimes performs worse than the no-subgoal ablation in *Length-Generalization* and *Switch-Concept-Order*, while ExeDec performs much better in *Compose-Different-Concepts* and *Compose-New-Operation*. For *Add-Operation-Functionality*, ExeDec and the no-subgoal ablation have a much smaller improvement over the Transformer baseline, especially in RobustFill. These observations may be explained by analyzing the different kinds of "spurious patterns" that arise in the various compositional generalization splits:

- For *Length-Generalization* and *Switch-Concept-Order*, the index of the current subprogram carries major implications in the training distribution, for example, "the problem should be almost solved by now" or "we must use this category of operation". However, these implications are drastically changed in the test distribution, leading to *spurious patterns* that may lead to poor performance on the test split. The Transformer baseline is aware of the length of the prediction so far, so it can be easily confused by this distribution shift — and indeed, it performs particularly poorly on these tasks. On the other hand, both ExeDec and the no-subgoal ablation are less aware of the current index of the subprogram, since the prior subprograms are only indirectly provided to the models through the program state. For these tasks, ExeDec and the no-subgoal ablation have relatively similar performance but greatly outperform the Transformer baseline.

- For *Compose-Different-Concepts* and *Compose-New-Operation*, the spurious patterns arise from comparison to what work needs to be done outside the current subprogram, for example, "this subprogram uses the same category of operation as the other subprograms" or "this subprogram can use operation $X$ only if there are no other subprograms". Again, these patterns are changed between the train and test distributions. The no-subgoal ablation is susceptible to overfitting on these patterns because it sees the I/O specification for the current subprogram composed with all future subprograms. On the other hand, ExeDec is more shielded from these spurious patterns because its SynthesizerModel only sees the I/O specification for the current subprogram (provided

that the SubgoalModel predicts the correct subgoals). This difference may explain why ExeDec has a larger performance improvement over the no-subgoal ablation for these generalization tasks.

- For *Add-Operation-Functionality*, the spurious pattern is actually within an individual subprogram, i.e., some subprograms in test problems are outside the distribution of subprograms seen during training. None of the compared approaches are well-shielded from this form of spurious pattern, leading to the relatively low performance of our methods on this generalization task for RobustFill. (For DeepCoder, the trend is less clear since some problems may be solved using longer programs outside the test distribution.)

## I    ALGORITHM COMPARISON CASE STUDY

**RobustFill.**    Figure 5 and Figure 6 compare how ExeDec, the no-subgoal ablation, and the Transformer baseline perform on two example problems from the RobustFill DSL, under the *Compose-New-Operation* generalization task where models were not trained on composed programs (other than length-1 programs) that use the `Compose` operation. A beam size of 1 (greedy decoding) is used for simplicity. In the figures, subprograms and execution results are colored red if they are incorrect. Portions of the final programs are colored yellow if they do execute correctly but implement the desired functionality suboptimally, i.e., with more subprograms than necessary.

For both example problems, ExeDec successfully solves the problem using a minimal-length program that is within the test distribution of the compositional generalization task, showing successful generalization. However, the no-subgoal ablation does not perform as well. In Figure 5, it uses the `Compose` operation incorrectly because, due to the *Compose-New-Operation* task, it has never been trained to use `Compose` to produce a *prefix* of the output. This is not an issue for ExeDec because the prefixes are provided by the SubgoalModel. In Figure 6, the ablation predicts a correct but suboptimal program in the training distribution, replacing a `Compose` operation with three separate steps. The Transformer baseline is comparatively the worst, predicting incorrect programs in the training distribution for both problems, showing a clear inability to compositionally generalize.

**DeepCoder.**    Figure 7 shows a similar comparison between the three approaches, this time on a DeepCoder list manipulation problem under the *Compose-Different-Concepts* generalization task. For this generalization task, we partition the DSL operations into two groups or "concepts" – one concept contains the higher-order operations `Scanl1`, `Filter`, `Count`, and `ZipWith`, while the other concept contains all of the first-order operations[6]. The training problems have minimal-length ground-truth programs that only compose operations within the same concept, while test problems require mixing operations from different concepts to achieve a minimal-length solution program.

The problem in Figure 7 is solved using operations from both concepts: `Scanl1` is higher-order, while `Take` and `Sort` are first-order operations. The Transformer baseline fails to solve this problem. It correctly uses `Scanl1` as the first step but incorrectly continues to use higher-order operations. This behavior makes sense because the model was trained on programs showing a similar pattern, and it does not deviate from the pattern because this approach is not compositionally general.

Similarly, the no-subgoal ablation correctly uses `Scanl1` in the first step but also continues to use higher-order operations in subsequent steps until eventually the program fails to typecheck. We observe that, after successfully using `Scanl1` in the first subprogram, the updated specification actually describes a problem with a solution program within the training distribution (using first-order operations `Take` and `Sort` from the same concept). So, why is the ablation unable to solve the updated specification, even though the SynthesizerModel is not directly conditioned on the previous subprograms like the Transformer baseline is? We hypothesize that the SynthesizerModel recognizes from the specification that `x2` was computed by applying a `Scanl1` operation to `x0`, and thus according to patterns seen during training, the model is inclined to continue using higher-order operations. This hypothesis is supported by the fact that the ablation, as well as ExeDec and the baseline, all correctly solve a simplified version of this problem where the input `x0` is replaced with the result of the first subprogram, `x2 = Scanl1 (+) x0`, such that the `Scanl1` portion is "already computed" but not in a way visible from the specification. In other words, the ablation succeeds for the simplified problem because it is now unable to refer to previous subprograms.

---

[6]The concept with first-order operations also contains the higher-order operation `Map`, to better balance the number of different program functionalities obtainable within each concept.

In contrast, ExeDec successfully solves the problem with a minimal-length solution that switches between concepts. Note that its SynthesizerModel cannot directly or indirectly reference previous subprograms. The SubgoalModel *can* indirectly refer to previous subprograms by examining the specification, but it is generally less susceptible to spurious compositional patterns. For example, the ablation's SynthesizerModel might learn the pattern "predict *an operation* in the same concept as previously" which is very easy for a neural model to recognize and implement, whereas it is more difficult for ExeDec's SubgoalModel to learn the pattern "predict *a subgoal that is implementable with an operation* in the same concept as previously". This provides some intuition for why ExeDec is more compositionally general overall.

## RobustFill Example Problem 1

**Specification:**
[ (*"alan Turing1"* → *"1.TURING,Alan"*),     (*"21.Donald@knuTh"* → *"21.KNUTH,Donald"*),
  (*"8:grace,HoppeR&"* → *"8.HOPPER,Grace"*), (*"EDSGER99 DIJKSTRA"* → *"99.DIJKSTRA,Edsger"*) ]

### ExeDec

**Step 1:**
- Subgoals: [*"1"*, *"21"*, *"8"*, *"99"*]
- Subprogram: `GetAll(NUMBER)`
- Execution: [*"1"*, *"21"*, *"8"*, *"99"*]
- Updated specification:
  [ (*"alan Turing1"* → *".TURING,Alan"*), **...** ]

**Step 2:**
- Subgoals: [*"."*, *"."*, *"."*, *"."*]
- Subprogram: `Const('.')`
- Execution: [*"."*, *"."*, *"."*, *"."*]
- Updated specification:
  [ (*"alan Turing1"* → *"TURING,Alan"*), **...** ]

**Step 3:**
- Subgoals: [*"TURING"*, *"KNUTH"*, *"HOPPER"*, **...** ]
- Subprogram: `Compose(ToCase(ALL_CAPS),`
                    `GetToken(WORD, -1))`
- Execution: [*"TURING"*, *"KNUTH"*, *"HOPPER"*, **...** ]
- Updated specification:
  [ (*"alan Turing1"* → *",Alan"*), **...** ]

**Step 4:**
- Subgoals: [*","*, *","*, *","*, *","*]
- Subprogram: `Const(',')`
- Execution: [*","*, *","*, *","*, *","*]
- Updated specification:
  [ (*"alan Turing1"* → *"Alan"*), **...** ]

**Step 5:**
- Subgoals: [*"Alan"*, *"Donald"*, *"Grace"*, **...** ]
- Subprogram: `Compose(ToCase(PROPER),`
                    `GetToken(WORD, 1))`
- Execution: [*"Alan"*, *"Donald"*, *"Grace"*, **...** ]
- Updated specification:
  [ (*"alan Turing1"* → *""*), **...** ]

No more steps: The updated specification output is empty for all examples.

Synthesis result: **Success**

Final program:
`GetAll(NUMBER) | Const('.') | Compose(ToCase(ALL_CAPS), GetToken(WORD, -1)) |`
`Const(',') | Compose(ToCase(PROPER), GetToken(WORD, 1))`

### No-Subgoal Ablation

**Step 1:**
- Subgoals: not predicted
- Subprogram: `GetToken(NUMBER, -1)`
- Execution: [*"1"*, *"21"*, *"8"*, *"99"*]
- Updated specification:
  [ (*"alan Turing1"* → *".TURING,Alan"*), **...** ]

**Step 2:**
- Subgoals: not predicted
- Subprogram: `Const('.')`
- Execution: [*"."*, *"."*, *"."*, *"."*]
- Updated specification:
  [ (*"alan Turing1"* → *"TURING,Alan"*), **...** ]

**Step 3:**
- Subgoals: not predicted
- Subprogram: `Compose(ToCase(ALL_CAPS),`
                    `GetFrom(','))`
- Execution: [*""*, *""*, *"HOPPER&"*, *""*]
- Updated specification: ***Invalid***

No more steps: In example #3, the subprogram's output *"HOPPER&"* is not a prefix of the desired output *"HOPPER,Grace"*.

Synthesis result: **Failure**

### Transformer Baseline

**Prediction:**
`GetToken(NUMBER, -1) | Const('.') |`
`ToCase(ALL_CAPS) | Const(',') |`
`Const('A') | GetFirst(LOWER, 1)`

Execution: [*"1.ALAN TURING1,Aalan"*,
          *"21.21.DONALD@KNUTH,Aonald"*,
          *"8.8:GRACE,HOPPER&,Agrace"*,
          *"99.EDSGER99 DIJKSTRA,A"*]

Synthesis result: **Failure**

Figure 5: A comparison of different approaches on the same string manipulation problem in the RobustFill domain, under the *Compose-New-Operation* generalization task. ExeDec is able to solve the problem correctly with a length 5 program including two usages of the new operation (`Compose`). The no-subgoal ablation fails to correctly use the `Compose` operation in step 3, likely because the model has not seen the `Compose` operation used to produce a *prefix* of the output. On the other hand, ExeDec succeeds in that step because the relevant prefixes are predicted as subgoals first. The Transformer baseline performs poorly on this task and does not use a single `Compose` operation.

## RobustFill Example Problem 2

**Specification:**
[ ("12.04 1999" → "1999/12/04"), ("07/08, 2000" → "2000/07/08"),
  ("04/12/1995" → "1995/04/12"), ("09.30.2001" → "2001/09/30") ]

### ExeDec

**Step 1:**
- Subgoals: ["1999", "2000", "1995", "2001"]
- Subprogram: GetToken(NUMBER, -1)
- Execution: ["1999", "2000", "1995", "2001"]
- Updated specification:
  [ ("12.04 1999" → "/12/04"), ... ]

**Step 2:**
- Subgoals: ["/", "/", "/", "/"]
- Subprogram: Const('/')
- Execution: ["/", "/", "/", "/"]
- Updated specification:
  [ ("12.04 1999" → "12/04"), ... ]

**Step 3:**
- Subgoals: ["12/04", "07/08", "04/12", ... ]
- Subprogram: Compose(Replace('.', '/'),
                       GetFirst(CHAR, 5))
- Execution: ["12/04", "07/08", "04/12", ... ]
- Updated specification:
  [ ("12.04 1999" → ""), ... ]

No more steps: The updated specification output is empty for all examples.

Synthesis result: **Success**

Final program:
GetToken(NUMBER, -1) | Const('/') |
Compose(Replace('.', '/'),
        GetFirst(CHAR, 5))

### Transformer Baseline

**Prediction:**
GetToken(NUMBER, -1) | Const('/') |
GetToken(DIGIT, 1) |
GetSpan(CHAR, 2, START, NUMBER, 1, END) |
Const('/') |
GetSpan(NUMBER, 1, START, CHAR, 5, END)

Execution: ["1999/12/12.04", "2000/07/07/08",
            "1995/04/04/12", "2001/09/09.30"]

Synthesis result: **Failure**

### No-Subgoal Ablation

**Step 1:**
- Subgoals: not predicted
- Subprogram: GetToken(NUMBER, -1)
- Execution: ["1999", "2000", "1995", "2001"]
- Updated specification:
  [ ("12.04 1999" → "/12/04"), ... ]

**Step 2:**
- Subgoals: not predicted
- Subprogram: Const('/')
- Execution: ["/", "/", "/", "/"]
- Updated specification:
  [ ("12.04 1999" → "12/04"), ... ]

**Step 3:**
- Subgoals: not predicted
- Subprogram: GetUpto(NUMBER)
- Execution: ["12", "07", "04", "09"]
- Updated specification:
  [ ("12.04 1999" → "/04"), ... ]

**Step 4:**
- Subgoals: not predicted
- Subprogram: Const('/')
- Execution: ["/", "/", "/", "/"]
- Updated specification:
  [ ("12.04 1999" → "04"), ... ]

**Step 5:**
- Subgoals: not predicted
- Subprogram: GetToken(NUMBER, -2)
- Execution: ["04", "08", "12", "30"]
- Updated specification:
  [ ("12.04 1999" → ""), ... ]

No more steps: The updated specification output is empty for all examples.

Synthesis result: **Success**

Final program:
GetToken(NUMBER, -1) | Const('/') |
GetUpto(NUMBER) | Const('/') |
GetToken(NUMBER, -2)

Figure 6: Another comparison on a different string manipulation problem. Here, ExeDec finds a minimal-length solution using 3 subprograms and 1 Compose operation. Meanwhile, the no-subgoal ablation solves the problem in a suboptimal way, replacing the Compose operation with 3 separate steps. The Transformer baseline also used a suboptimal approach in the middle of the program and was wrong in the last step.

### DeepCoder Example Problem

**Specification:**
```
[ ( { x0 = [5,-2,1],  x1 = 2 } → [3,5] ),  ( { x0 = [7,1,5,-2],  x1 = 4 } → [7,8,11,13] ),
  ( { x0 = [8,-4,-1,7],  x1 = 3 } → [3,4,8] ) ]
```

### ExeDec

**Step 1:**
- Subgoals: `[[5,3,4],[7,8,13,11],[8,4,3,10]]`
- Subprogram: `x2 = Scanl1 (+) x0`
- Execution: `[[5,3,4],[7,8,13,11],[8,4,3,10]]`
- Updated specification:
  `[ ( { x0 = [5,-2,1],  x1 = 2,  x2 = [5,3,4] }` `→ [3,5] ), … ]`

**Step 2:**
- Subgoals: `[[5,3],[7,8,13,11],[8,4,3]]`
- Subprogram: `x3 = Take x1 x2`
- Execution: `[[5,3],[7,8,13,11],[8,4,3]]`
- Updated specification:
  `[ ( { x0 = [5,-2,1],  x1 = 2,  x2 = [5,3,4],` `x3 = [5,3] } → [3,5] ), … ]`

**Step 3:**
- Subgoals: `[[3,5],[7,8,11,13],[3,4,8]]`
- Subprogram: `x4 = Sort x3`
- Execution: `[[3,5],[7,8,11,13],[3,4,8]]`
- Updated specification:
  `[ ( { x0 = [5,-2,1],  x1 = 2,  x2 = [5,3,4],` `x3 = [5,3],  x4 = [3,5] } → [3,5] ), … ]`

No more steps: The updated specification has `x4` matching the output for all examples.

Synthesis result: **Success**

Final program:
```
x0 = INPUT | x1 = INPUT |
x2 = Scanl1 (+) x0 | x3 = Take x1 x2 |
x4 = Sort x3
```

### Transformer Baseline

**Prediction:**
```
x0 = INPUT | x1 = INPUT |
x2 = Scanl1 (+) x0 |
x3 = Filter (%2==1) x2 |
x4 = ZipWith (min) x2 x3
```

Execution: `[[5,3],[7,8,11],[3]]`

Synthesis result: **Failure**

### No-Subgoal Ablation

**Step 1:**
- Subgoals: not predicted
- Subprogram: `x2 = Scanl1 (+) x0`
- Execution: `[[5,3,4],[7,8,13,11],[8,4,3,10]]`
- Updated specification:
  `[ ( { x0 = [5,-2,1],  x1 = 2,  x2 = [5,3,4] }` `→ [3,5] ), … ]`

**Step 2:**
- Subgoals: not predicted
- Subprogram: `x3 = Scanl1 (max) x2`
- Execution: `[[5,5,5],[7,8,13,13],[8,8,8,10]]`
- Updated specification:
  `[ ( { x0 = [5,-2,1],  x1 = 2,  x2 = [5,3,4],` `x3 = [5,5,5] } → [3,5] ), … ]`

**Step 3:**
- Subgoals: not predicted
- Subprogram: `x4 = Filter (>0) x0`
- Execution: `[[5,1],[7,1,5],[8,7]]`
- Updated specification:
  `[ ( { x0 = [5,-2,1],  x1 = 2,  x2 = [5,3,4],` `x3 = [5,5,5],  x4 = [5,1] } → [3,5] ), … ]`

**Step 4:**
- Subprogram: `x5 = Filter (%2==1) x2`
- Execution: `[[5,3],[7,13,11],[3]]`

**Step 5:**
- Subprogram: `x6 = Filter (>0) x3`
- Execution: `[[5,5,5],[7,8,13,13],[8,8,8,10]]`

**Step 6:**
- Subprogram: `x7 = ZipWith (min) x2 x6`
- Execution: `[[5,3,4],[7,8,13,11],[8,4,3,10]]`

**Step 7:**
- Subprogram: `x8 = Drop x2 x7`
- Execution: `TypeError`

No more steps: The last subprogram contains an error because `x2` is a list but `Drop` expects an integer.

Synthesis result: **Failure**

Figure 7: A comparison on a DeepCoder list manipulation problem under the *Compose-Different-Concepts* generalization task. All three approaches get the first step correct, but the ablation and baseline are unable to continue correctly, erroneously using higher-order operations for subsequent steps according to the compositional pattern in the training data. However, the design of ExeDec makes it less susceptible to overfitting on such patterns, and indeed ExeDec solves this problem using a minimal-length program from the test distribution.

# J  PROGRAMS AND PROMPTS FOR LLM EXPERIMENTS

## J.1  DSL PROGRAMS AS PYTHON FUNCTIONS

For the LLM experiments, we transform the DSL programs into Python functions that call a hypothetical `dsl` library, enabling the LLM to use its understanding of Python programming while we measure how well it generalizes to new functionality it has not been trained on.

The RobustFill program `GetFrom(' ') | Const('.') | Compose(ToCase(PROPER), GetToken(WORD, 1))` transforms the input string "TURING, Alan" into the output string "Alan.Turing". For the LLM experiments, it is written as a Python function as follows:

```python
def program(x):
  parts = [
      dsl.GetFrom(x, ' '),
      dsl.Const('.'),
      dsl.ToCase(dsl.GetToken(x, dsl.Type.WORD, 1), dsl.Case.PROPER),
  ]
  return ''.join(parts)
```

The DeepCoder program `x0 = INPUT | x1 = Map (**2) x0 | x2 = Sort x1` transforms the input list $[5, 3, -4]$ into the output list $[9, 16, 25]$. As a Python function, this would be:

```python
def program(x0):
  x1 = dsl.Map(dsl.SQUARE, x0)
  x2 = dsl.Sort(x1)
  return x2
```

We also experiment with a "Pythonic" form of DeepCoder programs, for example:

```python
def program(x0):
  x1 = [x ** 2 for x in x0]
  x2 = sorted(x1)
  return x2
```

## J.2  LLM PROMPTS

We programmatically create prompts for 3 LLM approaches (baseline, ablation, and ExeDec), for 3 kinds of datasets (RobustFill, DeepCoder, and DeepCoder-Pythonic). We provide prompts for a few example combinations in figures:

- Figure 8 has a Baseline-style prompt for RobustFill,
- Figure 9 has an Ablation-style prompt for DeepCoder,
- Figure 10 has an ExeDec-style prompt for RobustFill, and
- Figure 11 has an ExeDec-style prompt for DeepCoder-Pythonic.

In each figure, the prompt contains only 1 few-shot example for brevity, but our experiments used 4 few-shot examples for all prompts. Line wrapping is denoted with the $\hookrightarrow$ symbol.

Observe that the information contained in the ExeDec-style prompt is the same as in the Ablation-style prompt, just with different ordering of the code for a step and its corresponding execution results. Although this difference may seem slight, it leads to the improved performance for the ExeDec-style approach.

```
The `dsl` module is a custom library for manipulating strings. It
    ↪ contains the following functions:

Const, SubStr, GetSpan, GetToken, ToCase, Replace, Trim, GetUpto,
    ↪ GetFrom, GetFirst, GetAll, Substitute, SubstituteAll, Remove,
    ↪ RemoveAll

Additionally, the module defines the following constants:

dsl.Type.NUMBER, dsl.Type.WORD, dsl.Type.ALPHANUM, dsl.Type.ALL_CAPS,
    ↪ dsl.Type.PROP_CASE, dsl.Type.LOWER, dsl.Type.DIGIT,
    ↪ dsl.Type.CHAR, dsl.Case.PROPER, dsl.Case.ALL_CAPS,
    ↪ dsl.Case.LOWER, dsl.Boundary.START, dsl.Boundary.END

Below are example programming problems using the `dsl` module, with
    ↪ input-output test cases illustrating their behavior.

Important: All programs begin with ```python and end with ``` alone.

[BEGIN PROBLEM]
Input-output test cases:
  Case 1. "TURING, Alan" --> "Alan.Turing"
  Case 2. "knuth Donald" --> "Donald.Knuth"
  Case 3. "Hopper Grace" --> "Grace.Hopper"
  Case 4. "DIJKSTRA... Edsger" --> "Edsger.Dijkstra"

Program:
```python
def program(x):
  parts = [
      dsl.GetFrom(x, ' '),
      dsl.Const('.'),
      dsl.ToCase(dsl.GetToken(x, dsl.Type.WORD, 1), dsl.Case.PROPER),
  ]
  return ''.join(parts)
```
[END PROBLEM]

[BEGIN PROBLEM]
Input-output test cases:
  Case 1. "apple" --> "Apple!"
  Case 2. "banana" --> "Banana!"
  Case 3. "clementine" --> "Clementine!"
  Case 4. "durian" --> "Durian!"

Program:
```python
```

Figure 8: **Baseline-style prompt on RobustFill.** The LLM's continuation contains its predicted program as one entire function, terminated with triple backticks.

```
The `dsl` module is a custom library for manipulating lists of integers. It contains the
    ↪ following functions:

Head, Last, Take, Drop, Access, Minimum, Maximum, Reverse, Sort, Sum, Map, Filter, Count,
    ↪ ZipWith, Scan1l

Additionally, the module defines the following constants:

PLUS_ONE, MINUS_ONE, TIMES_TWO, DIV_TWO, NEGATE, SQUARE, TIMES_THREE, DIV_THREE,
    ↪ TIMES_FOUR, DIV_FOUR, IS_POSITIVE, IS_NEGATIVE, IS_EVEN, IS_ODD, ADD, SUBTRACT,
    ↪ MULTIPLY, MIN, MAX

Below are example programming problems using the `dsl` module, with input-output test cases
    ↪ illustrating the program behavior step-by-step.

Important: All programs begin with ```python and end with ``` alone.

[BEGIN PROBLEM]
Input-output test cases:
  Case 1. x0 = [5, 3, -4] --> [9, 16, 25]
  Case 2. x0 = [-2] --> [4]
  Case 3. x0 = [3, 7, 1, 4] --> [1, 9, 16, 49]

We solve this problem step-by-step.

Step 1 code:
```python
x1 = dsl.Map(dsl.SQUARE, x0)
```

Step 1 computes:
  Case 1. x1 = [25, 9, 16]
  Case 2. x1 = [4]
  Case 3. x1 = [9, 49, 1, 16]

Step 2 code:
```python
x2 = dsl.Sort(x1)
```

Step 2 computes:
  Case 1. x2 = [9, 16, 25]
  Case 2. x2 = [4]
  Case 3. x2 = [1, 9, 16, 49]

Putting the steps together, the problem is solved with the program:
```python
def program(x0):
  x1 = dsl.Map(dsl.SQUARE, x0)
  x2 = dsl.Sort(x1)
  return x2
```
[END PROBLEM]

[BEGIN PROBLEM]
Input-output test cases:
  Case 1. x0 = [1, 3, 5, 7], x1 = 2 --> [3, 9]
  Case 2. x0 = [2, -4, 1, 0, 5], x1 = 4 --> [6, -12, 3, 0]
  Case 3. x0 = [11], x1 = 3 --> [33]

We solve this problem step-by-step.

Step 1 code:
```

Figure 9: **Ablation-style prompt on DeepCoder.** For the ablation-style approach, the execution result of each step is shown *after* the code for that step. The LLM will respond with the first step toward solving the test problem, surrounded by triple backticks. We then execute that code (making some assumptions about its format, e.g., that it assigns to a variable) and construct a new prompt with the first step's code and execution results in the history, following the pattern in the few-shot example, so that the LLM's next prediction is the second step conditioned on the first step.

```
[... Description of the `dsl` module ...]

[BEGIN PROBLEM]
Input-output test cases:
  Case 1. x = "TURING, Alan" --> "Alan.Turing"
  Case 2. x = "knuth Donald" --> "Donald.Knuth"
  Case 3. x = "Hopper Grace" --> "Grace.Hopper"
  Case 4. x = "DIJKSTRA... Edsger" --> "Edsger.Dijkstra"

We solve this problem step-by-step.

Step 1 computes:
  Case 1. "Alan" so ".Turing" remains
  Case 2. "Donald" so ".Knuth" remains
  Case 3. "Grace" so ".Hopper" remains
  Case 4. "Edsger" so ".Dijkstra" remains

Step 1 code:
```python
dsl.GetFrom(x, ' ')
```

Step 2 computes:
  Case 1. "." so "Turing" remains
  Case 2. "." so "Knuth" remains
  Case 3. "." so "Hopper" remains
  Case 4. "." so "Dijkstra" remains

Step 2 code:
```python
dsl.Const('.')
```

Step 3 computes:
  Case 1. "Turing" so "" remains
  Case 2. "Knuth" so "" remains
  Case 3. "Hopper" so "" remains
  Case 4. "Dijkstra" so "" remains

Step 3 code:
```python
dsl.ToCase(dsl.GetToken(x, dsl.Type.WORD, 1), dsl.Case.PROPER)
```

Putting the steps together, the problem is solved with the program:
```python
def program(x):
  parts = [
      dsl.GetFrom(x, ' '),
      dsl.Const('.'),
      dsl.ToCase(dsl.GetToken(x, dsl.Type.WORD, 1), dsl.Case.PROPER),
  ]
  return ''.join(parts)
```
[END PROBLEM]

[BEGIN PROBLEM]
Input-output test cases:
  Case 1. x = "apple" --> "Apple!"
  Case 2. x = "banana" --> "Banana!"
  Case 3. x = "clementine" --> "Clementine!"
  Case 4. x = "durian" --> "Durian!"

We solve this problem step-by-step.

Step 1 computes:
```

Figure 10: **ExeDec-style prompt on RobustFill.** For the ExeDec-style approach, the execution result of a step comes *before* the code for that step. This is how we get the LLM to perform "execution decomposition" — it must predict execution subgoals before predicting code for that step. We extract the predicted code (surrounded by triple backticks), run it, and construct a new prompt containing this step's code and execution results (ignoring the model's predicted subgoals). The `dsl` module's description at the top of the prompt is omitted for space, but is identical to that in Figure 8.

```
The `dsl` module is a custom library for manipulating lists of integers. It contains the
     ↪ following functions:

def Scanl1(f, xs):
  ys = []
  for i, x in enumerate(xs):
    if i == 0:
      ys.append(x)
    else:
      ys.append(f(ys[-1], x))
  return ys

Below are example programming problems using the `dsl` module, with input-output test cases
     ↪ illustrating the program behavior step-by-step.

Important: All programs begin with ```python and end with ``` alone.

[BEGIN PROBLEM]
Input-output test cases:
  Case 1. x0 = [5, 3, -4] --> [9, 16, 25]
  Case 2. x0 = [-2] --> [4]
  Case 3. x0 = [3, 7, 1, 4] --> [1, 9, 16, 49]

We solve this problem step-by-step.

Step 1 computes:
  Case 1. x1 = [25, 9, 16]
  Case 2. x1 = [4]
  Case 3. x1 = [9, 49, 1, 16]

Step 1 code:
```python
x1 = [x ** 2 for x in x0]
```

Step 2 computes:
  Case 1. x2 = [9, 16, 25]
  Case 2. x2 = [4]
  Case 3. x2 = [1, 9, 16, 49]

Step 2 code:
```python
x2 = sorted(x1)
```

Putting the steps together, the problem is solved with the program:
```python
def program(x0):
  x1 = [x ** 2 for x in x0]
  x2 = sorted(x1)
  return x2
```
[END PROBLEM]

[BEGIN PROBLEM]
Input-output test cases:
  Case 1. x0 = [1, 3, 5, 7], x1 = 2 --> [3, 9]
  Case 2. x0 = [2, -4, 1, 0, 5], x1 = 4 --> [6, -12, 3, 0]
  Case 3. x0 = [11], x1 = 3 --> [33]

We solve this problem step-by-step.

Step 1 computes:
```

Figure 11: **ExeDec-style prompt on DeepCoder-Pythonic.** For DeepCoder-Pythonic, only the `Scanl1` operation remains in the hypothetical `dsl` library while every other operation and lambda function is written in Pythonic form. Thus, the beginning of the prompt has the library description updated correspondingly. We provide the implementation of `Scanl1` because it's only one function.

## K    FAILURE MODES IN LLM EXPERIMENTS

In the LLM experiments, ExeDec's incorrect solution programs encounter a variety of errors. Table 4 lists the specific errors, the number of programs encountering an error considering all generalization tasks (including no generalization), and the proportion of that error among all incorrect programs for that dataset.

Table 4: Error analysis for LLM experiments for ExeDec @ 1 (greedy decoding).

|  | RobustFill | | DeepCoder | | DeepCoder-Pythonic | |
|---|---|---|---|---|---|---|
| (Correct) | 33 | — | 66 | — | 88 | — |
| AssertionError | 2 | 0.2% | 0 | 0.0% | 0 | 0.0% |
| AttributeError | 6 | 0.5% | 63 | 5.6% | 0 | 0.0% |
| IndexError | 0 | 0.0% | 0 | 0.0% | 18 | 1.6% |
| NameError | 0 | 0.0% | 0 | 0.0% | 9 | 0.8% |
| SyntaxError | 1 | 0.1% | 1 | 0.1% | 0 | 0.0% |
| TypeError | 234 | 20.1% | 150 | 13.2% | 9 | 0.8% |
| ValueError | 0 | 0.0% | 0 | 0.0% | 3 | 0.3% |
| ZeroDivisionError | 0 | 0.0% | 0 | 0.0% | 15 | 1.3% |
| Wrong behavior | 924 | 79.2% | 920 | 81.1% | 1058 | 95.1% |

For RobustFill, about 20% of failures are `TypeErrors` caused by using the DSL incorrectly, while about 79% of failures do not encounter a runtime error but simply have behavior inconsistent with the I/O specification.

For DeepCoder, `AttributeError` accounts for about 5.6% of failures (when the predicted program attempts to use a hallucinated function or constant in the DSL), `TypeError` accounts for about 13% of failures, and about 81% of failures are due to wrong behavior. For DeepCoder-Pythonic, about 5% of failures are from various runtime errors, while about 95% are due to wrong behavior.

Overall, the vast majority of failures are due to wrong behavior. Although LLMs have seen much success in predicting programs from natural language specifications, they still perform poorly in programming-by-example without natural language hints, especially for synthetic problems.

## L    HIERARCHICAL EXEDEC

The following is Python-like pseudocode describing how ExeDec might be extended to enable hierarchical decompositions. We leave exploration of this extension to future work.

```
def ExeDecHierarchical(inputs, outputs):
  difficulty = DifficultyModel(inputs, outputs)  # outputs easy/hard
  if difficulty == 'easy':  # base case
    return SynthesizerModel(inputs, outputs)

  # recursive case
  subprograms = []
  while True:
    subgoals = SubgoalModel(inputs, outputs)
    subprogram = ExeDecHierarchical(inputs, subgoals)  # recurse
    subprograms.append(subprogram)
    execution_result = Execute(subprogram, inputs)
    if execution_result == outputs:
      return CombineProgramParts(subprograms)
    (inputs, outputs) = UpdateSpecification(inputs, outputs,
                                            execution_result)
```

