# OpenReview forum: "ExeDec: Execution Decomposition for Compositional Generalization in Neural Program Synthesis"
_ICLR.cc/2024/Conference — ICLR 2024 oral_

### Official Review · Reviewer_EB19 · 2023-10-26

**Soundness:** 3 good
**Presentation:** 2 fair
**Contribution:** 3 good
**Rating:** 6
**Confidence:** 4

**Summary:**

The paper proposes a set of generalization tasks to measure the ability of a programming by example program synthesis engine to generalize to be able to generate out-of-distribution programs. These tasks measure the ability of the synthesis engine to generalize to programs of different lengths and with different operations in different orderings than observed in the training data. The paper then proposes ExeDec, an iterative synthesis approach in which one model proposes a subgoal, a different model proposes a program to each that subgoal, and the process repeats until the composed program results in the expected output. The paper evaluates ExeDec instantiated with Transformers trained for each generalization task and with LLMs and find that ExeDec improves generalization compared to baselines.

**Strengths:**

* The proposed set of generalization tasks are interesting, important, and well described
* The proposed ExeDec algorithm is also interesting: in particular, it is a general algorithm sketch that can be instantiated either by Transformers trained for this algorithm (Sections 4.2/5.1) or LLMs (Section 5.2)
* The results, taken at face value (though see the question about FLOPs below) are strong. In particular, they demonstrate that the core idea of predicting the next output improves generalization.

**Weaknesses:**

* The benchmark introduced in Sections 2 and 3 is claimed as a novel contribution ("we introduce a new meta-benchmark for measuring the compositional generalization abilities of program synthesizers."). However, it is very hard to tell from these descriptions the actual contents and novelty of the benchmark. I was ultimately able to get a better understanding from reading the evaluation and Appendix B, but in isolation the clarity of these sections are very low.
* The implications of the loop in Algorithm 1 are not sufficiently discussed:
  * What happens if the loop never terminates?
  * What is the comparison in FLOPs between the ExeDec models and the baseline models (including the beam search, etc)? Even if the parameter count is the same, the loop in ExeDec could give these models significantly more power than the Transformer / Latent Programmer baselines.

**Questions:**

* What happens if the loop in Algorithm 1 never terminates?
* My reading of the results in 5.2 (specifically, Table 1) is that I should divide all numbers by 200 to compare them with the results in 5.1 (Figure 2). Is this correct?
* What is the comparison in FLOPs between the ExeDec models and the baseline models?
* How do models perform on generalization tasks that they were not trained for?

My score is conditioned on the authors improving the description of the baseline in Sections 2 and 3.

---

> ### Author Response · Authors · 2023-11-17
> **Response to Reviewer EB19**
>
> Thank you for your review! We are grateful for the opportunity to work with you to improve our paper.
>
> > The benchmark introduced in Sections 2 and 3 is claimed as a novel contribution ... However, it is very hard to tell from these descriptions the actual contents and novelty of the benchmark. I was ultimately able to get a better understanding from reading the evaluation and Appendix B, but in isolation the clarity of these sections are very low.
>
> Thank you for the feedback. We are glad to hear that you were able to get a better understanding from reading the evaluation and Appendix B, not that information was missing entirely. Perhaps some rearranging of information could be done to improve the flow.
>
> To help us take action on this, could you please let us know what information from the evaluation and Appendix B was most helpful to improving your understanding, so we can move that information to Sections 2 and 3? It would also be super helpful to hear suggestions for what other information in the paper is not as important and can be moved away to an appendix, to make space.
>
> Please keep in mind that we are already at the limit of 9 pages exactly, we will have even less space in the camera ready (due to the author list and no additional pages allowed), and we tried our best to partition the most-information in the main text from the less-important information in the appendices (14 additional pages of information).
>
> > What happens if the loop in Algorithm 1 never terminates?
>
> In our experiments, we impose a maximum number of steps, after which we declare that the method fails to solve that problem. This is described in Appendix C, where the beam search checks for beam states where “some computation limit such as a maximum number of steps is reached”.
>
> For the models trained from scratch:
> * For RobustFill, we have a limit of 20 steps, but we almost never reach that limit because we also require that each subprogram makes correct progress with respect to the desired output. Most of the time, if a subprogram prediction is wrong, incorrect progress will be made (which we cannot recover from due to the concatenation structure of RobustFill programs) and the beam state will be marked as invalid and removed from the beam search.
> * For DeepCoder, we have a limit of using 10 variables in the entire program (including variables used for the inputs), where each step uses a fresh variable.
>
> For the LLM experiments: we use a limit of 3 steps.
>
> > My reading of the results in 5.2 (specifically, Table 1) is that I should divide all numbers by 200 to compare them with the results in 5.1 (Figure 2). Is this correct?
>
> In Table 1, dividing the numbers by 200 results in percentages of problems solved. However, those percentages are not comparable with those in Figure 2 because the problems in the LLM experiments are easier, as explained in Section 5.2 and Footnote 4.
>
> > What is the comparison in FLOPs between the ExeDec models and the baseline models?
>
> The loop in our step-by-step synthesis approaches does not lead to more “power”, because our step-by-step approaches predict multiple shorter subprograms, while the Transformer baseline and Latent Programmer predict the entire long program at once. If we assume that subprograms, a set of subgoals for each example, and I/O specifications all have an equal number of tokens, then ExeDec-Small and the No-Subgoal Ablation use roughly the same number of FLOPs, or slightly more FLOPs than the Transformer baseline and Latent Programmer (only because those approaches don’t need to encode the step-by-step execution information).
>
> > How do models perform on generalization tasks that they were not trained for?
>
> The best way to answer this question may be to use the models trained without generalization, tested on the various generalization datasets. Our results for “test on training distribution” (i.e., the no-generalization case) can be seen as a combination of this for all generalization tasks, since the no-generalization distribution contains all of the generalization train and test distributions. But, the exact test problems themselves are different. We will investigate this and include the numbers in the revised paper.

---

### Official Review · Reviewer_iTFM · 2023-10-31

**Soundness:** 3 good
**Presentation:** 3 good
**Contribution:** 3 good
**Rating:** 8
**Confidence:** 3

**Summary:**

The paper proposes a method for programming by examples where the goal is to generate a program for given input-output examples. The proposed method first predicts subgoals (which are simply intermediate values during the program execution) and then generates a (sub)program that achieves those subgoals. This is repeated until the subgoals correspond to the target outputs. Additionally, the paper suggests different benchmarks for compositional generalization for programming by examples (e.g., generalizing to longer programs). The proposed method is shown to have better generalization performance than the baselines in both the training from scratch and the pre-trained setting.

**Strengths:**

The proposed method for programming mirrors how humans tend to come up with programs for complicated programming tasks: divide them into smaller subproblems and address them individually. In that sense, the method is simple and intuitive, hence I'd expect something similar to work well even in more realistic settings.

The benchmarks provide valuable insights into the settings under which we would expect a decomposition into subprograms to be important.

It is interesting to see that the proposed method can also be adapted to work for pre-trained LLMs in the few-shot setting (i.e., without fine-tuning).

**Weaknesses:**

The main limitations are that the method requires supervision for the sub-goals and that the experiments are limited to synthetic programs, as acknowledged by the authors in the limitations section. I can see that an unsupervised approach to predicting subgoals constitutes its own paper, but I think it'd still be interesting to see how the method fares with different decompositions. For example, picking $n$ lines per subprogram, or randomly grouping multiple lines into a subprogram.

The experiments cover both the training from scratch scenario and a pre-trained LLM. The middle ground of finetuning a pre-trained LLM would be interesting to see. Here the same LLM could be used for the different models via different prompts (similar to the few-shot setting) and trained like the from-scratch setting. Since the benchmark intends to measure generalization and considering the importance of pre-training on the generalization performance, I believe this is an important additional experiment.

ExeDec requires subgoals which are essentially execution traces as supervision. It'd be interesting to see how the Transformer baseline performs when it also has access to them (for example by inserting the variable values after/before each program line). This baseline would further clarify the importance of training with intermediate program states versus decomposing the program into parts that are individually generated.

**Questions:**

Do you have an insight into why the no-subgoal ablation performs better on some generalization benchmarks?

---

> ### Author Response · Authors · 2023-11-17
> **Response to Reviewer iTFM**
>
> Thank you for your review!
>
> > The middle ground of finetuning a pre-trained LLM would be interesting to see.
>
> This would be interesting but we figured that, if we had to fine-tune the LLM 18 times (3 DSL settings, 6 generalization settings), this experiment would become prohibitively expensive.
>
> You might be suggesting that we fine-tune the LLM 3 times, once for each of the DSL settings (RobustFill, DeepCoder, and DeepCoder-Pythonic), using the training distribution from the no-generalization setting. This is still expensive, and we thought this experimental setup would lead to less interpretable results, since we’d be testing on a distribution of programs that were seen during fine-tuning but are absent from the few-shot examples. We also weren’t sure how such a setup might mirror a real-life scenario, as typically one does not have a large finetuning dataset comprising all of the generalization patterns seen at test time.
>
> > It'd be interesting to see how the Transformer baseline performs when it also has access to [execution traces]
>
> This is also an interesting suggestion. We assume you mean that we should “pause” the Transformer baseline’s program prediction after each line, collect the execution trace, inject the trace into the decoding, and then “unpause” the prediction to get the next line. As a step-by-step program generation procedure with program execution between steps, this is already quite similar to the No-Subgoal Ablation.
>
> If we then consider that the Transformer baseline has no way of “backtracking” to change its prior prediction, and we already have a setup where we can specify all remaining work for the problem in the form of I/O examples, it would make sense that each step should not care about what prior predictions or execution traces led to the current state. In fact, conditioning on the prior steps could actually be distracting to the model, teaching it about spurious patterns that harm the generalization performance. If we treat each subprogram as a fresh prediction that is not conditioned on the previous steps, we arrive (at least in spirit) to the No-Subgoal Ablation.
>
> > Do you have an insight into why the no-subgoal ablation performs better on some generalization benchmarks?
>
> This may be related to the kind of spurious pattern involved in the compositional generalization split. Specifically:
>
> * For Length-Generalization and Switch-Concept-Order, the index of the current subprogram carries major implications in the training distribution, i.e., “the problem should be almost solved by now” or “we must use this category of operation”. However, those implications are drastically changed in the test distribution. The Transformer baseline is aware of the length of the prediction so far, so it can be easily confused by the distribution shift -- and indeed, it performs particularly poorly on these tasks. On the other hand, both ExeDec and the No-Subgoal Ablation are less aware of the current index of the subprogram, since the prior subprograms are only indirectly provided to the models through the program state. For these tasks, the No-Subgoal Ablation sometimes outperforms ExeDec slightly.
> * For Compose-Different-Concepts and Compose-New-Operation, the spurious pattern arises from comparison to what work needs to be done outside the current subprogram: “this subprogram uses the same category of operation as the other subprograms” or “this subprogram can only use operation X if there are no other subprograms”. Again, these patterns are changed between the train and test distributions. The No-Subgoal Ablation is susceptible to overfitting on these patterns because it sees the I/O specification for the current subprogram composed with all future subprograms. On the other hand, ExeDec is shielded from these spurious patterns because its SynthesizerModel only sees the I/O specification for the current subprogram (provided that the SubgoalModel predicts the correct subgoals), which may explain the larger performance gap.
> * For Add-Operation-Functionality, actually the spurious pattern is within an individual subprogram, i.e., some subprograms in test problems are outside the distribution of subprograms seen during training. None of the compared approaches are well-shielded from this form of spurious pattern, leading to the relatively low performance of our methods on this generalization task for RobustFill. (For DeepCoder, the trend is less clear since some problems may be solved using longer programs outside the test distribution.)
>
> We’d be happy to add this discussion to the paper. Thanks for bringing it up! (Reviewer hCQX also asked a similar question, and the discussion above is copy/pasted in our responses to both of you.)

---

### Official Review · Reviewer_hCQX · 2023-11-03

**Soundness:** 3 good
**Presentation:** 3 good
**Contribution:** 3 good
**Rating:** 8
**Confidence:** 3

**Summary:**

Summary:
- This work formalizes several useful notions of generalization, and describes how these can be used to create train-test splits of datasets testing each form of generalization. The classic neural synthesis datasets DeepCoder and RobustFill  are divided up in this way and used for evaluation throughout the paper.
- The authors then presents a novel architecture called ExeDec for synthesizing sequential programs, where a prefix of a partially constructed program can be executed to get a state-so-far. While prior work conditions next-subprogram generation on the state-so-far and overall output spec, here the authors introduce a subgoal-generating module that produces an output spec for just the next subprogram, then conditions next-subprogram generation on that spec (though regardless of whether the subgoal spec is met, the sampled subprogram is accepted).
- While performing relatively similar to baselines/ablations on the original datasets, ExeDec performs far better than alternatives at the generalization-focused modified datasets. Additionally, in another set of experiments the authors prompt an LLM to perform ExeDec-style reasoning (predicting outputs of the next line before writing it) and find it boosts performance.

**Strengths:**

- Formalizing different notions of generalization and suggesting actionable ways of designing benchmarks around this is much-needed work in program synthesis. This is something that I come back to again and again in my own thinking – the scarcity of generalization-centric benchmarks and systematic generalization evaluations, especially beyond just length generalization.

- Evaluations look quite good – ExeDec Performance is far better than the from-scratch transformer and latent programmer baselines, and barely diminishes when using the smaller version of the model.

- The improvements over the No-Subgoal ablation (which is similar to prior work as mentioned in 4.3) are fairly significant (eg 80% -> 87%) and are mainly in the Compose New Operation and Compose Different Concept categories. These aren't extremely strong results but they're enough to justify the relatively straightforward general setup, in my view.

- Showing off and evaluating an analogous algorithm for LLM-based synthesis was a nice touch and shows the generality of the idea. This idea of breaking problems into subgoals is a good one, and I think one that many have thought about but nobody has done well – this work feels like a nice step towards doing it. As discussed in their limitations, something more hierarchical might better capture how programmers often break down problems, however just predicting the results of the next line is a reasonable first step in this direction and seems to empirically be helpful.
    - In fact, before reading this I actually would not have expected predicting the output of just the next subprogram to be that helpful compared to just predicting the subprogram directly (whereas I would expect predicting outputs to more general hierarchical multi-subprogram subproblems to be helpful) so I appreciate that this paper finds and highlights where even just this single step prediction can be beneficial.

**Weaknesses:**

- In the intro there's a mention of how compositional generalization "has not previously been studied in the context of programming by example" and later in the related work a mention of how there's "less work on systematic generalization for machine learning for code, although Bieber et al. (2020) studies length generalization"
    - While both systematic generalization and specifically compositional generalization are understudied in program synthesis, there are a few other citations that apply to each of these places.
    - First, the DeepCoder paper (Balog 2017) actually evaluates on length generalization (section 5.2 of that paper).
    - Nye et al 2021 ("Representing Partial Programs with Blended Abstract Semantics") has some compositional generalization evaluation as well, such as the excerpt from section 5.1 "Tower objects seen during training were composed in previously unseen ways..."
    - Ellis et al 2019 Program Synthesis in a REPL (already cited here) also applies for length generalization, eg the line from 4.2 "Although it was trained on programs whose maximum length was 30 actions and average length approximately 8 actions, during test time we regularly achieved programs with 40 actions or more"
    - Just adding those citations in the appropriate places would strengthen this. To be clear, ExeDec's formalization of generalization and range of types of generalization go far beyond any of this prior work, but it's important to acknowledge that there have been some explorations of some of these kinds of generalization in the past.
    - Aside from this point, I think the related work citations are quite thorough and well done

- How often are proposed subgoals actually achieved? Are they achieved more along a successful synthesis path than an unsuccessful one, on average? Some quantitative insights into these sorts of questions would strengthen the paper.

- The results are decent but not extremely strong – in my opinion they're good enough but of course stronger results would help.

- The idea mentioning in the Limitations section of how subgoals that are not just line-by-line but which could instead correspond to more than one line is one that immediately came to mind. Of course, fully generally doing that kind of generic hierarchical decomposition of a problem into subproblems is something of a holy grail in synthesis, and I think that this fairly limited form of subproblem is a reasonable step, so this is not a huge weakness.

**Questions:**

- In Figure 2, why do you think that ExeDec improves over the No-Subgoal ablation primarily in the Compose New Operation and Compose Different Concepts settings, and not in the other settings? Some added discussion of this would be helpful.

- "Details omitted for anonymity. This LLM is one of the largest available through an API." I don't think anonymizing the LLM if it's available though a public API (eg chatgpt, bard, claude, codex, gpt4, etc) is necessary, and none of the other papers I'm reviewing do this. It'd be helpful to know which LLM this is (unless it's a private API and would actually break anonymity of the authors) in case it raises any other questions/discussion from reviewers.

- I'm a little surprised by how poorly the LLMs do in Table 1. Do you have a sense for the usual failure modes – does it produce code that's valid in terms of the DSL, but isnt correct? Or does it fail to adhere to the DSL? (as an aside, in the non-pythonic setup, if the LLM attempts to write normal python code instead of using the `dsl` library, is this treated as invalid and do you have a sense for how often this happens?)

- To clarify, "test on training distribution" in Fig 2 is referring to a heldout test set of problems from the same distribution as the training set, and not testing directly on any of the problems that the models were trained on, right?

---

> ### Author Response · Authors · 2023-11-17
> **Response to Reviewer hCQX**
>
> Thank you for the in-depth review and your encouraging comments! Due to the character limit per message, we respond in two messages.
>
> > Just adding those citations in the appropriate places would strengthen this.
>
> You’re absolutely right, we will revise the text and add the citations. Thank you for suggesting this!
>
> > How often are proposed subgoals actually achieved?
>
> We measured a related metric of “single-step accuracy” for the SubgoalModel and SynthesizerModel. That is, for a single step in a trajectory that matches the ground-truth so far, how often does the SubgoalModel predict subgoals exactly matching the ground-truth ones (for all I/O examples) with a single greedy decoding? And, how often does the SynthesizerModel predict a program whose behavior matches the ground-truth subgoal?
>
> For RobustFill, the SubgoalModel gets 90% - 98% single-step accuracy for the different generalization tasks, while the SynthesizerModel gets 84% on Add-Operation-Functionality and 94% - 99% on the other tasks.
>
> For DeepCoder, the SubgoalModel gets 56% accuracy for no-generalization and 16% - 41% on the generalization tasks, while the SynthesizerModel gets 99% on no-generalization, 60% on Add-Operation-Functionality, 78% on Switch-Concept-Order, and 91 - 96% on the other tasks.
>
> Note that our “single-step accuracy” metric is particularly low for the SubgoalModel on DeepCoder because there are potentially many correct ways of solving the problem, and this metric only considers the single ground-truth solution. In fact, the SubgoalModel does not need to have super high accuracy in order to achieve good end-to-end results, because the SynthesizerModel can ignore slight errors in the subgoals and still produce a program that behaves as closely as possible while only using 1 DSL operation. We have seen many concrete cases of the SynthesizerModel being given a slightly-incorrect subgoal and then producing the correct subprogram anyway, which disagrees with the subgoal but correctly makes progress overall.
>
> > In Figure 2, why do you think that ExeDec improves over the No-Subgoal ablation primarily in the Compose New Operation and Compose Different Concepts settings, and not in the other settings?
>
> This may be related to the kind of spurious pattern involved in the compositional generalization split. Specifically:
>
> * For Length-Generalization and Switch-Concept-Order, the index of the current subprogram carries major implications in the training distribution, i.e., “the problem should be almost solved by now” or “we must use this category of operation”. However, those implications are drastically changed in the test distribution. The Transformer baseline is aware of the length of the prediction so far, so it can be easily confused by the distribution shift -- and indeed, it performs particularly poorly on these tasks. On the other hand, both ExeDec and the No-Subgoal Ablation are less aware of the current index of the subprogram, since the prior subprograms are only indirectly provided to the models through the program state. For these tasks, the No-Subgoal Ablation sometimes outperforms ExeDec slightly.
> * For Compose-Different-Concepts and Compose-New-Operation, the spurious pattern arises from comparison to what work needs to be done outside the current subprogram: “this subprogram uses the same category of operation as the other subprograms” or “this subprogram can only use operation X if there are no other subprograms”. Again, these patterns are changed between the train and test distributions. The No-Subgoal Ablation is susceptible to overfitting on these patterns because it sees the I/O specification for the current subprogram composed with all future subprograms. On the other hand, ExeDec is shielded from these spurious patterns because its SynthesizerModel only sees the I/O specification for the current subprogram (provided that the SubgoalModel predicts the correct subgoals), which may explain the larger performance gap.
> * For Add-Operation-Functionality, actually the spurious pattern is within an individual subprogram, i.e., some subprograms in test problems are outside the distribution of subprograms seen during training. None of the compared approaches are well-shielded from this form of spurious pattern, leading to the relatively low performance of our methods on this generalization task for RobustFill. (For DeepCoder, the trend is less clear since some problems may be solved using longer programs outside the test distribution.)
>
> We’d be happy to add this discussion to the paper. Thanks for bringing it up! (Reviewer iTFM also asked a similar question, and the discussion above is copy/pasted in our responses to both of you.)

---

> > ### Author Response · Authors · 2023-11-17
> > **Response to Reviewer hCQX, continued**
> >
> > > It'd be helpful to know which LLM this is
> >
> > The LLM we used is actually only available to certain organizations at the moment. If you like, we can share more specific details with the AC who may decide how to proceed.
> >
> > > Do you have a sense for the usual failure modes [in the LLM experiments?]
> >
> > * In about 0 - 10% of failures (the exact proportion heavily depends on the dataset and generalization setting), the predicted program attempts to use a hallucinated function or constant in the DSL, leading to an AttributeError.
> > * About 0 - 2% of failures are from a SyntaxError.
> > * About 5 - 20% of failures are from some other runtime error during program execution, most often a TypeError caused by using the DSL in the wrong way.
> > * The most common failure mode by far, in about 75 - 95% of failures, is when the predicted program executes without error but does not adhere to the I/O examples. In our experience (in other projects as well), LLMs are quite bad at performing programming-by-example without natural language specifications.
> >
> > > in the non-pythonic setup, if the LLM attempts to write normal python code instead of using the dsl library, is this treated as invalid and do you have a sense for how often this happens?
> >
> > We do not require the LLM to use the `dsl` module in any setting; any Python code is allowed as long as it meets formatting requirements expected by our step-by-step synthesis procedure.
> >
> > > To clarify, "test on training distribution" in Fig 2 is referring to a heldout test set of problems from the same distribution as the training set, and not testing directly on any of the problems that the models were trained on, right?
> >
> > The training and test datasets were generated independently, so by random chance, there may be some overlap in the ground-truth programs (especially for short ones), although having exact match on the I/O examples as well would be very unlikely.

---

> > > ### Comment · Reviewer_hCQX · 2023-11-22
> > >
> > > Thanks for the responses – I'll be keeping my score of an 8 supporting the acceptance of the paper.
> > >
> > > By the way, it doesn't seem like you've uploaded a revision of the paper, but I trust that the stated changes will be made. Please be sure to include this "single-step accuracy" analysis in the final version (main paper or appendix).
> > >
> > > > single-step accuracy
> > >
> > > This is interesting! Thank you. I do think the other related analysis I mentioned would also be helpful, especially when it comes to cases like you mention where there's more than one way of solving a problem (so you want to see how well the synthesizer accomplishes the proposed subgoal instead of the ground truth). However I'm certainly not demanding those in this revision – just an idea for future analyses you might be interested in.
> > >
> > > > For Compose-Different-Concepts and Compose-New-Operation, the spurious pattern arises from... ExeDec is shielded from these spurious patterns because its SynthesizerModel only sees the I/O specification for the current subprogram
> > >
> > > Hm right, the SubgoalModel still seems like it needs to deal with the spurious pattern here maybe though – but I can sortof see how that might be easier. And in practice it does seem like it makes it work better.
> > >
> > > > Spurious patterns... We’d be happy to add this discussion to the paper.
> > >
> > > That would be great – yes the analysis of these 3 spurious patterns in the final version of the paper (as some speculative intuition for what might be happening) would be helpful to readers / the community I think! As is it felt like the results on the different splits were presented but not really the intuition for why they might be how they are. I think this pattern analysis would be great.
> > >
> > > > In about 0 - 10% of failures ...
> > >
> > > These quick stats on failure modes might be nice to include somewhere as well, even if it's just an appendix.
> > >
> > >
> > > > We do not require the LLM to use the `dsl` module in any setting; any Python code is allowed ...
> > >
> > > Ahh – could you add this to the paper as a note if it's not in there already? I was thinking we were still ultimately trying to fit even the pythonic versions into a more traditional strict DSL program synthesis setup, I didn't realize generic python was allowed. That works though, just clarify it in the paper.
> > >
> > > > The training and test datasets were generated independently, so by random chance, there may be some overlap in the ground-truth programs (especially for short ones), although having exact match on the I/O examples as well would be very unlikely.
> > >
> > > Gotcha. In general would be good to make these strictly disjoint sets, but given that most of the focus of this paper is on the generalization splits anyways this isn't a huge deal.
> > >
> > > Thanks for the discussion and clarification on all my questions!

---

> > > > ### Author Response · Authors · 2023-11-23
> > > > **Thank you!**
> > > >
> > > > Thank you again for your review and discussion! We will definitely include these extra analyses and clarifications in the final version.

---

### Official Review · Reviewer_2WRW · 2023-11-03

**Soundness:** 3 good
**Presentation:** 4 excellent
**Contribution:** 2 fair
**Rating:** 6
**Confidence:** 4

**Summary:**

This paper improves the composition generalization of neural program synthesis by introducing execution decomposition. The general idea is to interleave sub-goal predictions with program generation. To evaluate this idea, the authors design five compositional generalization tasks based on two popular datasets used by RobustFill and DeepCoder. All specifications including sub-goals are assumed to be I/O-example based, where the inputs of sub-goals are intermediate program states. The outputs of sub-goals are somewhat complicated depending on the application domain. The evaluation shows that ExeDec outperforms many other baselines that do not (explicitly) predict sub-goals, and LLMs struggle to solve the proposed tasks requiring compositional generalization even when ExeDec-style prompts are provided.

**Strengths:**

- Having compositional generalization as an inductive bias in the synthesis algorithm is novel and sounds quite appealing. Compositional generalization should be a key aspect in order to achieve scalable program synthesis.
- Five different types of compositional generalization tasks are designed and relevant benchmarks are created based on two popular synthesis domains, RobutFill and DeepCoder.
- The evaluation consists of multiple baselines as well as LLMs, and ExeDec outperforms all of them on the proposed dataset
- The problem is well-motivated and the related work is discussed in an insightful and thorough manner

**Weaknesses:**

- The two datasets are small domain-specific tasks, thus the evaluation does not really show the promise of scalability, which is the original motivation.
- I/O-based specifications are not easy to construct -- both intermediate states and output states may vary significantly from the initial given I/O examples. For instance, intermediate states may involve auxiliary temporary variables, and the output states may have to be adjusted according to the actual application domains.
- Both DSLs seem simple straight-line code, and there are no recursion, loops, or if-else conditions, all of which are important components to construct a realistic large program.
- Creating a good training dataset for decomposition tasks can be very challenging. Particularly, there are many valid solutions, which may have quite different intermediate sub-goals. Supervising only one specific solution would be problematic.

**Questions:**

How to process different numbers of I/O examples (the size of which may also vary) in transformer models?

Straight-line code can be split arbitrarily, how to decide which snippets form a proper sub-goal?

It is a bit surprising that No-subgoal Ablation performs pretty well. Shouldn't it behave like the Transformer baseline?

---

> ### Author Response · Authors · 2023-11-17
> **Response to Reviewer 2WRW**
>
> Thank you for your review and thoughtful questions and comments!
>
> > I/O-based specifications are not easy to construct
>
> This is certainly true in general, but I/O specifications are still a common and intuitive way of specifying programs. Indeed, in unit tests, educational or competitive programming challenges, and on forums like StackOverflow, I/O examples are often expected. Of course, they may be combined with other forms of specification like natural language, but developing approaches that reason about I/O examples is still an important aspect of program synthesis overall.
>
> > Both DSLs seem simple straight-line code, and there are no recursion, loops, or if-else conditions, all of which are important components to construct a realistic large program.
>
> In general-purpose programming languages, these control flow constructs are definitely important. Extending the ExeDec approach to handle these is important future work. That said, we note that many important programming libraries do not rely on these explicit control flow constructs. For example, matrix or tensor manipulation with TensorFlow / PyTorch / Numpy for machine learning and scientific computing, exploratory data analysis with Pandas, plotting with Matplotlib, spreadsheet formulas in Excel, and MapReduce pipelines are all typically expressed with a sequence or composition of library operations, usually without explicit control flow structures from the underlying programming language. In this sense, developing techniques that work well for straight-line code is still an important research direction.
>
> > Particularly, there are many valid solutions, which may have quite different intermediate sub-goals. Supervising only one specific solution would be problematic.
>
> If there are multiple routes to a solution, in theory we would like to teach the model about all “good” (e.g., minimal-weight) solutions. In a hypothetical infinite-training-data regime, this is equivalent to selecting a random “good” solution independently for each training example. Then in practice, with some care in implementation, one can approximate this ideal by choosing uniformly random minimal-weight solutions when generating fresh random training data for every training step. (Our code does not yet select among minimal-weight solutions uniformly, but we do generate fresh training data for each training step.)
>
> > How to process different numbers of I/O examples (the size of which may also vary) in transformer models?
>
> In our model (Section 4.2), a Transformer encoder turns each I/O example $X_i$ into an encoding $\\phi_i$, which has fixed length regardless of the example’s size. To combine these $\\phi_i$ across examples, we simply concatenate them which does assume (as in our experiments) that every problem is specified with a constant number of examples.
>
> More generally, to handle any number of I/O examples, one can instead pool across examples, $\\phi_{\\text{pool}} \\gets \\operatorname{Pool}(\\{\\phi_i\\})$. Then, when predicting a subgoal for example $X_i$, the SubgoalModel may use a concatenation of $\\phi_{\\text{pool}}$ (encoding the overall problem) and $\\phi_i$ (encoding the specific example for which we need a subgoal), passed to a Transformer decoder which predicts the desired subgoal. The SynthesizerModel can directly apply a Transformer decoder to $\\phi_{\\text{pool}}$ to predict a subprogram.
>
> > Straight-line code can be split arbitrarily, how to decide which snippets form a proper sub-goal?
>
> In our experiments, we make initial progress by simply using a subgoal after every line (with one DSL operation per line). Future work may investigate our ideas on more realistic code, where there are other signals that may identify plausible subgoals. For instance, blank lines, comments, and logging statements often break up real code into logical steps which we can use as subgoals. Even if the ground-truth code does not contain any such signals, one may prompt a code language model to guess where blank lines might be added, to receive a suggestion based on the language model’s ability to pattern-match to other code. Applying these ideas to real code would be very exciting to explore in future work, made possible by our initial findings on synthetic code.
>
> > It is a bit surprising that No-subgoal Ablation performs pretty well. Shouldn't it behave like the Transformer baseline?
>
> There are two main difference between these approaches. First (and most importantly), the No-Subgoal Ablation receives program execution information after every predicted subprogram, so it can change its solution on the fly depending on the current program state, while the Transformer Baseline receives no execution information at all. Second, the No-Subgoal Ablation is trained on individual subprograms, while the Transformer Baseline was not explicitly shown how entire programs are composed of individual subprograms. In our view, these differences make the relative comparison expected.

---

### Meta-Review · Area_Chair_ZJyn · 2023-12-06

**Metareview:**

I recommend the acceptance of the paper based on the reviews and responses provided by the authors. The key reasons for acceptance are:

Novelty and Importance: The paper presents a novel approach to programming by example, introducing the concept of execution decomposition and subgoal prediction. This method mirrors human problem-solving strategies and is a significant step towards more intuitive program synthesis.

Comprehensive Evaluation: The evaluation of the proposed method, ExeDec, is thorough, encompassing multiple baselines and large language models (LLMs). The method demonstrates superior performance in compositional generalization tasks, highlighting its effectiveness in a crucial aspect of program synthesis.

Benchmarks for Generalization: The introduction of new benchmarks for measuring compositional generalization abilities in program synthesizers is a notable contribution. These benchmarks address a critical gap in program synthesis research, providing valuable tools for future work in this area.

Adaptability to LLMs: The adaptability of ExeDec for use with pre-trained LLMs in a few-shot setting, without fine-tuning, is particularly impressive. This flexibility demonstrates the potential applicability of the approach in various settings and with different underlying technologies.

Responses to Reviewers: The authors have addressed the concerns and questions raised by the reviewers comprehensively and satisfactorily. Their commitment to include additional analyses and clarifications in the final version of the paper is commendable and enhances the paper's quality.

**Justification For Why Not Higher Score:**

N/A

**Justification For Why Not Lower Score:**

The research goes beyond specific technical contributions, addressing broader questions of how to approach program synthesis more effectively and intuitively. This has significant implications for the field of AI and programming.

While there are limitations, such as the need for supervised sub-goals and the focus on synthetic programs, the authors acknowledge these and propose them as directions for future research. The potential for finetuning the approach in more realistic settings and with different decompositions offers exciting avenues for further exploration.

---

### Decision · Program_Chairs · 2024-01-16

Accept (oral)